# Rad-NeRF: Ray-decoupled Training of Neural Radiance Field

**Lidong Guo**[1*]     **Xuefei Ning**[1*†]     **Yonggan Fu**[2]     **Tianchen Zhao**[1]

**Zhuoliang Kang**[3]     **Jincheng Yu**[1]     **Yingyan (Celine) Lin**[2]     **Yu Wang**[1†]

[1]Tsinghua University     [2]Georgia Institute of Technology     [3]Meituan

## Abstract

Although the neural radiance field (NeRF) exhibits high-fidelity visualization on the rendering task, it still suffers from rendering defects, especially in complex scenes. In this paper, we delve into the reason for the unsatisfactory performance and conjecture that it comes from interference in the training process. Due to occlusions in complex scenes, a 3D point may be invisible to some rays. On such a point, training with those rays that do not contain valid information about the point might interfere with the NeRF training. Based on the above intuition, we decouple the training process of NeRF in the ray dimension softly and propose a **Ray-decoupled** Training Framework for neural rendering **(Rad-NeRF)**. Specifically, we construct an ensemble of sub-NeRFs and train a soft gate module to assign the gating scores to these sub-NeRFs based on specific rays. The gate module is jointly optimized with the sub-NeRF ensemble to learn the preference of sub-NeRFs for different rays automatically. Furthermore, we introduce depth-based mutual learning to enhance the rendering consistency among multiple sub-NeRFs and mitigate the depth ambiguity. Experiments on five datasets demonstrate that Rad-NeRF can enhance the rendering performance across a wide range of scene types compared with existing single-NeRF and multi-NeRF methods. With only 0.2% extra parameters, Rad-NeRF improves rendering performance by up to 1.5dB. Code is available at https://github.com/thu-nics/Rad-NeRF.

## 1 Introduction

Novel view synthesis is an important task within the domains of computer vision and computer graphics, playing an essential role in a variety of applications, such as autonomous driving, augmented reality, and so on. Recently, Neural Radiance Field (NeRF) [17] has emerged as a promising solution, achieving high-fidelity visualizations on the novel view synthesis task. It implicitly encodes 3D scenes through neural networks and trains the networks using volume rendering.

Despite NeRF's excellent scene representation ability, it still suffers from rendering defects when dealing with complex scenes, such as 360-degree unbounded scenes [37, 2] and large scenes with free shooting trajectories [30, 27, 26]. One of the main reasons is the limited model capacity. However, directly increasing the network's size yields marginal performance improvement [18].

**Our fundamental intuition is that the training interference from invisible rays affects NeRF's performance.** Let us consider a simple case of a 360-degree unbounded scene with a central object

---

[*]Both authors contribute equally to this work.

[†]Corresponding authors: Xuefei Ning (foxdoraame@gmail.com), Yu Wang (yu-wang@tsinghua.edu.cn).

38th Conference on Neural Information Processing Systems (NeurIPS 2024).

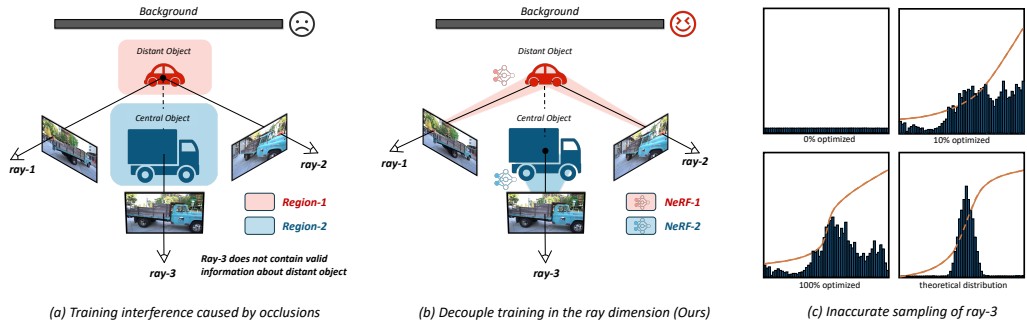

| (a) Training interference caused by occlusions | (b) Decouple training in the ray dimension (Ours) | (c) Inaccurate sampling of ray-3 |

Figure 1: A case in 360-degree unbounded scenes (bird-eye view). (a) For the distant object, invisible ray-3 interferes with ray-1/2 training. (b) The ray-based multi-NeRF framework considers variable visibility of objects to different rays and decouples training in the ray dimension. (c) Compared to the theoretical weight distribution, the sampling along ray-3 is inaccurate incurring training interference.

(truck) and a distant object (car). As illustrated in Figure 1(a), a 3D point located on the distant object can be observed from ray-1 and ray-2, but is invisible to ray-3 due to the occlusion presented by the central object. Although NeRF models transmittance in its volume rendering formula, it exhibits low geometric modeling accuracy and inaccurate sampling distribution in complex scenes, especially at the start of training, as Figure 1(c) shows. So, 3D points on the distant object might be sampled by the ray-3, and the model is trained on these points by the ray-3 color. However, ray-3 does not contain any meaningful information about the distant object, potentially interfering with the NeRF's training. In contrast, considering the different visibility of the object to different rays, our intuition is that rather than using one NeRF, assigning the rays terminating at the distant object to NeRF-1 and the rays terminating at the central object to NeRF-2 could be better, as shown in Figure 1(b).

To verify the above intuition, we manually select two sets of images in the TAT dataset[13]. One set contains 80 images of the train's front side, while the other set includes the former set and 80 backside images. We train two NeRFs using these two sets respectively. As shown in Figure 2, the model trained on the mixed set performs worse on the front side, which matches our intuition.

To mitigate the training interference caused by invisible rays, the intuition solution is to decouple the training of the rays terminating at different regions. To this end, we propose a **ray-decoupled training framework for neural rendering (Rad-NeRF)**. Within the Rad-NeRF framework, an ensemble of sub-NeRFs has different preferences for different rays through a gate module. With the help of the gate module, sub-NeRFs' outputs are fused by post-volume-rendering fusion to yield final rendering results. Notably, the gate module is jointly optimized with NeRF, allowing it to automatically learn the preference of each sub-NeRF for various rays in an end-to-end manner. This **learnable gating** design makes Rad-NeRF generally applicable to diverse scenes, which stands in contrast to prior multi-NeRF methods [27, 26] that rely on manually defined allocation rules.

Additionally, we design a depth-based mutual learning method for the multi-NeRF framework to ensure the rendering consistency among multiple sub-NeRFs. In addition to learning colors, sub-NeRFs teach each other with their rendered depths. Traditional NeRF methods may struggle with generalization to novel views despite accurately rendering training views, as they often fail to capture precise geometry [7, 37]. In contrast, our depth-based mutual learning approach serves as a form of geometric regularization, alleviating the depth ambiguity and avoiding overfitting.

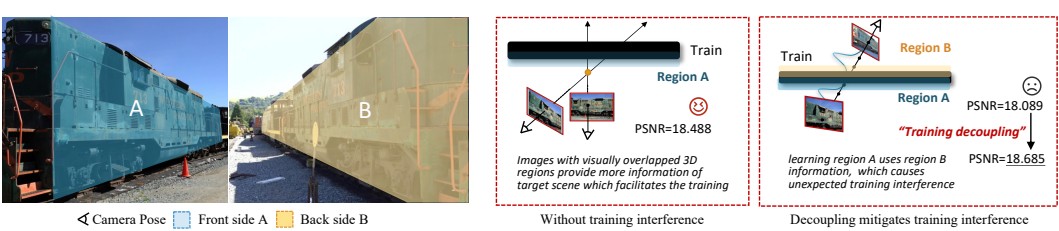

| ◁ Camera Pose ▢ Front side A ▢ Back side B | Without training interference | Decoupling mitigates training interference |

Figure 2: Oracle experiment: Training interference from invisible rays affects NeRF's performance.

To verify the effectiveness of Rad-NeRF, we conduct extensive experiments on various types of datasets. The results show that Rad-NeRF can exhibit *anti-aliasing effects* and obtain *superior geometry modeling*, thus consistently improving the rendering quality of novel views. In addition, Rad-NeRF is *parameter-efficient* and *super simple to implement*. With only 0.2% extra parameters, Rad-NeRF can increase rendering performance by up to 1.5dB compared to Instant-NGP. By scaling the number of sub-NeRFs through ray-wise decoupling, Rad-NeRF achieves better performance-to-parameter scalability than scaling other dimensions, such as the MLP width or the feature grid.

## 2 Related Work

### 2.1 Neural Radiance Field

Neural Radiance Field (NeRF) [17] has received much attention since it was proposed. It uses MLPs to implicitly represent 3D objects or scenes, achieving realistic rendering results. There have been intensive studies on NeRF's extension, including increasing NeRF's training/inference efficiency [36, 8, 21, 25, 18, 5], applying NeRF to specific scenes (large/unbounded/poor-textured) [15, 37, 2, 31, 26, 27, 38], applying NeRF to other tasks (surface reconstruction/scene editing) [35, 20, 29, 14, 33, 32], increasing NeRF rendering quality in few-shot setting [10, 12, 19, 7]. In this work, we aim to increase NeRF's rendering quality in complex scenes, and propose a multi-NeRF training framework, which can leverage the techniques proposed by these single-NeRF researches.

### 2.2 Multi-NeRF Representation

Due to the limited model capacity, the multi-NeRF method is widely adopted to improve the rendering quality, which can be categorized into point- and ray-based multi-NeRF methods.

**Point-based multi-NeRF method.** These methods divide the 3D space in the point-dimension [30, 37, 38]. 3D points in different regions are computed by different sub-NeRFs. For example, NeRF++ [37] proposes the sphere inversion transformation to map an infinite space to a bounded sphere first, and it uses two NeRFs to model the foreground and background regions respectively. Switch-NeRF [38] also partitions the scenes in the point-dimension. These methods do not consider the different visibility of a target region to different views and cause training interference on complex scenes with many occlusions. For example, the front side of an object is not visible when it is observed from the back view or blocked by an occlusion. Training the sub-NeRF with rays that do not contain any valid information about the target region might interfere with the training.

**Ray-based multi-NeRF methods.** These methods allocate training rays to different sub-NeRFs and train sub-NeRFs independently. Block-NeRF [26] and Mega-NeRF [27] perform the ray allocation in the image-granularity and pixel-granularity, respectively. Both of them need a manually defined allocation rule, which requires prior scene knowledge and cannot be easily adapted to other types of scenes. The former work trains sub-NeRFs in large-scale road scenes with prior knowledge of the image shooting position distribution, and the latter one trains sub-NeRFs in open drone scenes and allocates the rays based on the ray intersecting positions with a horizontal plane. However, defining a ray allocation rule for complex scenes lacking prior scene-specific knowledge remains challenging. Another related work is NID [28], which proposes a mixture-of-experts NeRF for generalizable scene modeling. In this work, different experts serve as the basis to construct the implicit field of different scenes and the gating module takes in the new scene's image as the input (i.e., image-granularity).

In this work, we propose a gate-guided multi-NeRF mutual learning framework, performing the allocation and decoupling the training in the ray dimension softly. Compared to other multi-NeRF methods, Rad-NeRF boosts the rendering quality without the need for prior scene knowledge.

## 3 Preliminary

NeRF [17] uses neural networks to represent 3D scenes implicitly. Two MLPs model the density and color of spatial points respectively. The input of density MLP $F_\sigma$ is the 3D point coordinate $\mathbf{x}$. The input of color MLP $F_c$ includes view direction $\theta$ and feature $f$ output by density MLP. NeRF proposes the volume rendering method to render each pixel of an image. It samples $N$ points along

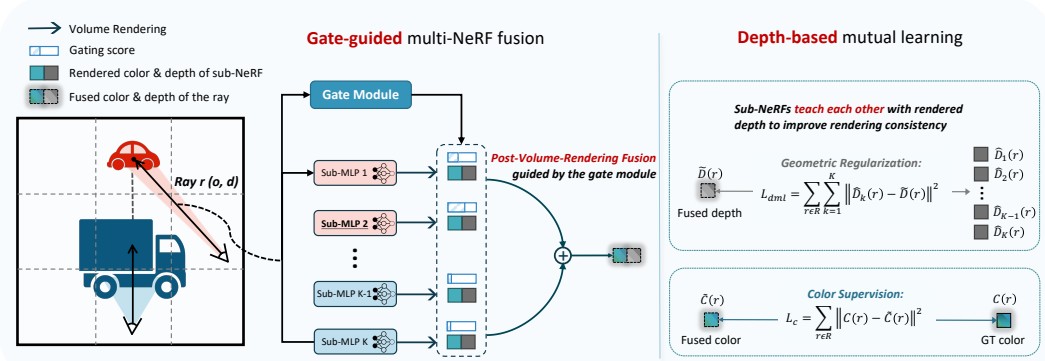

Figure 3: The overview of Rad-NeRF. We construct a multi-NeRF framework based on the hybrid representation, where the feature grid is shared for all sub-NeRFs and the MLP decoders are independent. **(Left)** Given a ray, the soft gate module encodes the ray's data and outputs a soft score. Then, guided by the gating score, sub-NeRFs' outputs are fused after the volume rendering process. **(Right)** The fused rendered depth of the ray is used to regularize each sub-NeRF's geometric encoding.

the ray and renders the pixel's color $\hat{C}(\mathbf{r})$ by discretely summing density $\sigma_i$ and color $\mathbf{c}_i$ of each point $i$, which approximates the integral $C(\mathbf{r})$ as follows:

$$C(\mathbf{r}) = \int_0^{+\infty} w(t)\mathbf{c}(t)dt \quad \hat{C}(\mathbf{r}) = \sum_{i=1}^{N} w_i \mathbf{c}_i, \tag{1}$$

$$T_i = \exp\left(-\sum_{j=1}^{i-1} \sigma_j \delta_j\right) \quad w_i = T_i\left(1 - e^{-\delta_i \sigma_i}\right), \tag{2}$$

where $t_i$ is the distance between $i$-th sample's position and the starting point of the ray, $\delta_i = t_{i+1} - t_i$ is the distance between adjacent samples and $T_i$ represents the probability that the ray travels from the start to point $i$ without hitting. The NeRF optimization is based on color supervision.

## 4 Rad-NeRF

NeRF faces the challenge of limited model capacity when rendering complex scenes [37, 30, 38]. However, directly increasing the number of model parameters yields marginal improvement in the rendering quality [18], posing an important research question: "how to effectively scale up the capacity of NeRF". While the multi-NeRF methods have been proposed as an effective technique in response to this question, they still face limitations in handling complex scenes (with many occlusions and arbitrary shooting trajectories) due to training interference among invisible rays. In this work, we propose a ray-decoupled training framework (Rad-NeRF), effectively scaling up model's capacity by decoupling training in the ray dimension in a learnable way. Figure 3 gives an overview of Rad-NeRF.

### 4.1 Gate-guided Multi-NeRF Fusion

Motivated by the intuition and oracle experiment discussed in Section 1, rather than using a single NeRF model, designing a multi-NeRF structure that considers different visibility of the region to different rays and decouple NeRF's training in the ray-dimension could be better. We design a ray-based multi-NeRF model structure and introduce a soft gate module to learn the preference of each sub-NeRF for various rays in a learnable way.

**Multi-NeRF Structure.** As shown in Figure 3, we employ a shared feature grid among sub-NeRFs and keep MLP decoders independent for the multi-NeRF structure. As different rays may pass through the same region of 3D space, weight sharing for the feature grid helps training, owing to the feature grid's responsibility for encoding features of 3D spatial points. As validated by the Oracle experiment, training with regions A and B facilitates the training of the shared feature grid

and improves the rendering quality(PSNR 18.685 vs 18.488). Meanwhile, as the MLP decoder is designed to encode view information, constructing an ensemble of independent MLP decoders helps to decouple the training in the ray dimension, and thus maintains the preference of sub-NeRFs for various rays. Additionally, such structure is a multi-model extension of Instant-NGP [18], helping to avoid a significant increase in the number of parameters and training complexity. The hybrid representation also maintains high training efficiency.

**Soft Gate Module.** We incorporate a soft gate module to assign gating scores to the sub-NeRFs for each ray. The soft gate module is jointly optimized with NeRF, enabling it to learn the preference of each sub-NeRF for different rays in an end-to-end manner. In contrast to manually assigning training rays to sub-NeRFs, this learnable gating design makes Rad-NeRF **generally applicable to diverse scenes lacking prior scene-specific knowledge**. In Section 5.2, we will also show that the gate module can learn to assign reasonable gating scores that reflect the object visibility of rays, aligning with our intuition that decoupling training in the ray-dimension is important.

Specifically, we employ a four-layer MLP followed by a Softmax function as the gate module. The gate module takes the starting point and direction $(o,d)$ of a ray $\mathbf{r}$ as the input, and outputs the gating scores $\boldsymbol{G}(\mathbf{r})$ of multiple sub-NeRFs associated with this ray. Instead of applying any sparsification strategies on the gating score $\boldsymbol{G}(\mathbf{r})$ as in previous work [38], such as top-k gating function [23], we use soft gating scores to enhance the smoothness and consistency of rendered results.

As discussed in Section 3, each ray corresponds to a pixel on the image. Following the volume rendering process, we can obtain $K$ rendered colors for each ray, where $K$ is the number of sub-NeRFs. Subsequently, multi-NeRFs' outputs are fused in a post-volume-rendering ordering to obtain the final rendering results. The fused color $\tilde{C}(\mathbf{r})$ of the ray $\mathbf{r}$ can be written as below:

$$\tilde{C}(\mathbf{r}) = \sum_{k=1}^{K} G_k(\mathbf{r})\hat{C}_k(\mathbf{r}), \tag{3}$$

where $G_k(\mathbf{r})$ is the $k$-th element of gating score $\boldsymbol{G}(\mathbf{r})$ and $\hat{C}_k(\mathbf{r})$ is the rendered color of $k$-th sub-NeRF for the ray $\mathbf{r}$.

## 4.2 Depth-based Mutual Learning

By the learnable soft gating design, different sub-NeRFs learn different encodings of the scene. We introduce a mutual learning method to enhance the rendering consistency and robustness of sub-NeRFs, wherein each sub-NeRF not only learns from ground truth but also learns from each other. Due to the lack of the ground truth for per-ray depth, NeRF may fail to learn accurate geometry despite accurately rendering training views, which adversely affects its generalization to novel views. To address this, we perform mutual learning with the rendered depths of sub-NeRFs, which serves as a form of geometric regularization and helps the model find more robust geometric solutions. The per-ray depth estimation $\hat{D}(\mathbf{r})$ can be written as Equation 4, where $t_i$ is the i-th sample's distance from the starting point on the ray.

$$\hat{D}(\mathbf{r}) = \sum_{i=1}^{N} w_i t_i, \tag{4}$$

In practice, we first fuse the rendered depths of sub-NeRFs guided by the gating score $\boldsymbol{G}(\mathbf{r})$. Then we use L2 distance to quantify the match of each sub-NeRF's rendered depth $\hat{D}_k(\mathbf{r})$ and the fused depth $\tilde{D}(\mathbf{r})$. Our depth-based mutual learning loss is defined as below, where $\mathcal{R}$ is the set of sampled rays:

$$L_{dml} = \sum_{\mathbf{r}\in\mathcal{R}} \sum_{k=1}^{n} \|\hat{D}_k(\mathbf{r}) - \tilde{D}(\mathbf{r})\|^2, \tag{5}$$

Compared to directly averaging multiple sub-NeRFs' depth predictions, the gate-guided fused depth $\tilde{D}(\mathbf{r})$ is more accurate, as the gating score $\boldsymbol{G}(\mathbf{r})$ can reflect the prediction confidence of each sub-NeRF for the ray $\mathbf{r}$.

Table 1: Quantitative results in complex scenes.

| Methods | TAT | | | NeRF-360-v2 | | | Free-Dataset | | |
|---|---|---|---|---|---|---|---|---|---|
| | PSNR↑ | SSIM↑ | LPIPS↓ | PSNR↑ | SSIM↑ | LPIPS↓ | PSNR↑ | SSIM↑ | LPIPS↓ |
| NeRF++ | 20.419 | 0.663 | 0.451 | 27.211 | 0.728 | 0.344 | 24.592 | 0.648 | 0.467 |
| MipNeRF360 | **22.061** | **0.731** | **0.357** | **28.727** | **0.799** | **0.255** | **27.008** | **0.766** | 0.295 |
| MipNeRF360$_{short}$* | 20.078 | 0.617 | 0.508 | 25.484 | 0.631 | 0.452 | 24.711 | 0.648 | 0.466 |
| DVGO | 19.750 | 0.634 | 0.498 | 25.543 | 0.679 | 0.380 | 23.485 | 0.633 | 0.479 |
| Instant-NGP | 20.722 | 0.657 | 0.417 | 27.309 | 0.756 | 0.316 | 25.951 | 0.711 | 0.312 |
| F2-NeRF | – | – | – | 26.393 | 0.746 | 0.361 | 26.320 | **0.779** | **0.276** |
| Switch-NGP† | 20.512 | 0.654 | 0.432 | 26.524 | 0.740 | 0.331 | 25.755 | 0.694 | 0.341 |
| Block-NGP† | 20.783 | 0.659 | 0.415 | 27.436 | 0.761 | 0.298 | 26.015 | 0.702 | 0.325 |
| Rad-NeRF | **21.708** | **0.672** | **0.398** | **27.871** | **0.769** | **0.298** | **26.449** | 0.719 | **0.285** |

\* MipNeRF360 requires nearly one day for training. For a fair comparison, we also report its results with one-hour of training.
† We adapt Switch-NeRF and Block-NeRF to the Instant-NGP fast training framework.

### 4.3 The Overall Training Loss

The overall loss function of Rad-NeRF is given by:

$$L = L_c + \lambda_1 L_{dml} + \lambda_2 L_{cv}, \tag{6}$$

where $L_c = \sum_{\mathbf{r} \in \mathcal{R}} \|C(\mathbf{r}) - \tilde{C}(\mathbf{r})\|^2$ ($C(\mathbf{r})$ is the ground truth color value of ray $\mathbf{r}$) is the rendering loss. $\lambda_1$ and $\lambda_2$ are the weights for regularization terms, which are the only hyper-parameters to be set. The value of $\lambda_1$ is chosen from $1 \times 10^{-4}$ and $5 \times 10^{-3}$. $\lambda_2$ is set to $1 \times 10^{-2}$ on all the datasets. $L_{cv}$ is the balancing regularization on the Coefficient of Variation of the soft gating scores, which prevents the gate module from collapsing onto a specific sub-NeRF. The details of $L_{cv}$ are described and discussed in the Appendix B.

## 5 Experiments

### 5.1 Datasets and Baselines

**Datasets.** We use five datasets from different types of scenes to evaluate our Rad-NeRF. (1) Object dataset: we take *Masked Tanks-And-Temples dataset (MaskTAT)* [13] for evaluation, which are photographed objects with masked background; (2) 360-degree inward/outward-facing datasets: we take *Tanks-And-Temples (TAT) dataset* with unmasked background [13] and *NeRF-360-v2 dataset* [2] to evaluate on scenes with large dynamic depth range; (3) free shooting-trajectory datasets: we conduct experiments on *Free-Dataset [30]* and *ScanNet dataset* [6], which are large outdoor and indoor scenes respectively. Both larger view ranges and more irregular shooting trajectories pose greater challenges for NeRF rendering.

**Baselines.** We compare our Rad-NeRF with two types of methods: one type uses the grid-based NeRF framework as we do, including PlenOctrees [36], DVGO [25], Instant-NGP [18] and F2-NeRF [30]. The other one is the MLP-based NeRF method, including NeRF [17], NeRF++ [37], MipNeRF [1] and MipNeRF360 [2], which is inefficient in training and needs almost one day for training in complex scenes. Note that we also implement the NGP-version of Block-NeRF [26], Switch-NeRF [38] and Mega-NeRF [27] to validate the superiority of Rad-NeRF to other multi-NeRF methods. The implementation details of Mega-NGP are shown in the Appendix F.

### 5.2 Comparative Studies

**Rad-NeRF achieves higher rendering quality than existing single- and multi-NeRF methods.** We report the main quantitative results on the complex scenes and the object dataset in Table 1 and Appendix D respectively. Within no more than one hour of training, Rad-NeRF achieves higher rendering quality compared to other fast training methods and multi-NeRF methods, including Switch-NGP and Block-NGP. We can also see that while Rad-NeRF is designed for complex scene

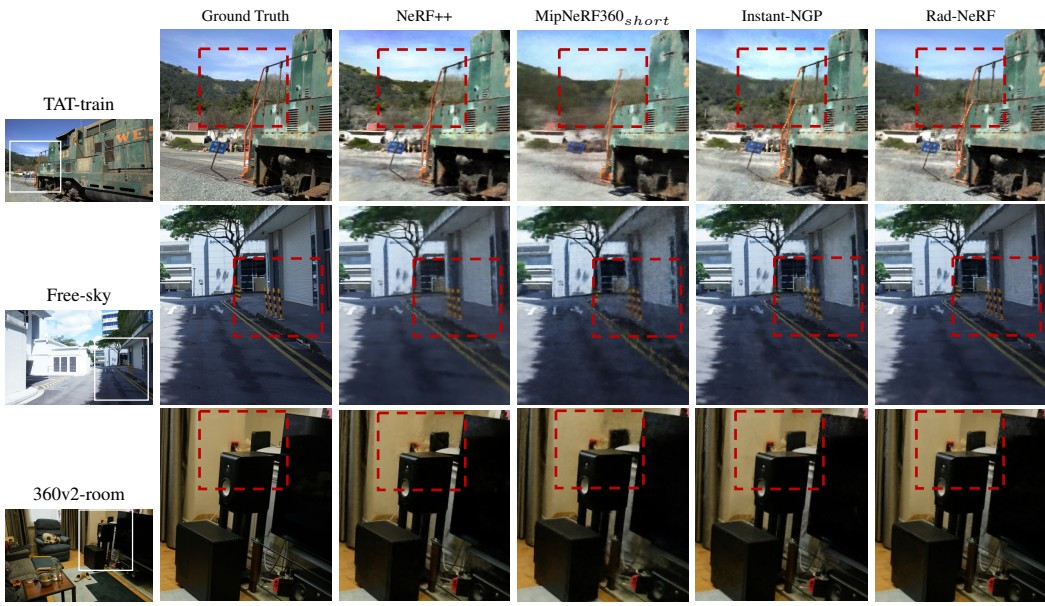

Figure 4: Qualitative comparisons on three complex scenes. Rad-NeRF achieves better recovery of details for distant objects and less textured regions such as the wall. (Zoom in for the details, e.g., sky, banister, roadblock, wall.)

rendering, it can also improve the rendering performance of objects. We also integrate Rad-NeRF with the recent SOTA single-NeRF framework ZipNeRF [3], named Rad-ZipNeRF, in the Appendix I. Rad-ZipNeRF obtains better rendering performance, validating Rad-NeRF's potential for integration with different frameworks.

**Rad-NeRF achieves better recovery of distant details and accurate rendering for less textured regions.** The qualitative results are shown in Figure 4. Compared to other methods, Rad-NeRF achieves better rendering quality in both outdoor and indoor scenes. In outdoor scenes, Rad-NeRF produces detailed and realistic rendering results for the sky and other distant objects. In indoor scenes, Rad-NeRF generates more accurate details for less textured regions such as the wall. Rad-NeRF takes advantage of the gate-guided training decoupling in the ray dimension to boost the model's performance effectively. Results on the ScanNet dataset are shown in the Appendix C.

**The gate module learns to reasonably assign gating scores.** We visualize how the gate module performs training decoupling in Figure 5. As the two sub-NeRFs exhibit complementary gating scores, we omit sub-NeRF2's visualization for brevity. (1) In the Truck scene, the gate module assigns different preferences to sub-NeRF1 in foreground/background regions, thereby mitigating the interference from foreground rays on sub-NeRF1's training with the background region. (2) In the Train scene, sub-NeRF1 exhibits higher preferences for the back side, thereby mitigating the

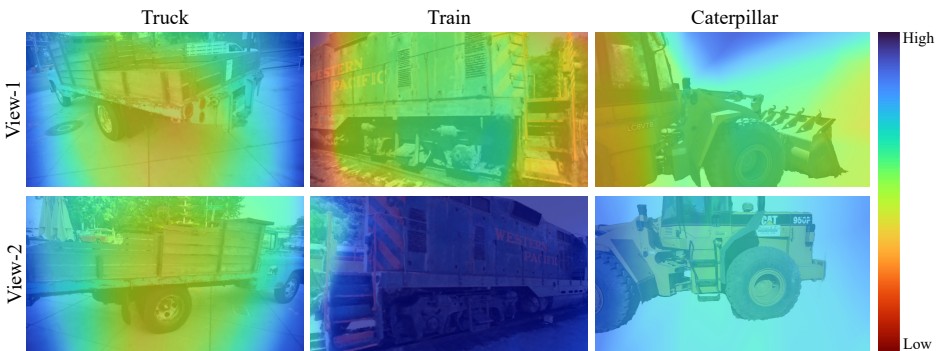

Figure 5: Visualization of the gating scores of sub-NeRF1 on two different views (visualization of sub-NeRF2 is omitted for brevity).

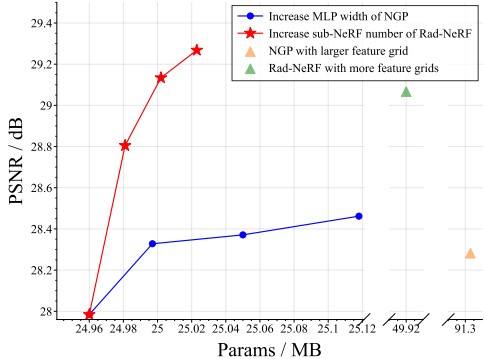

Figure 6: Scalability study of Rad-NeRF.

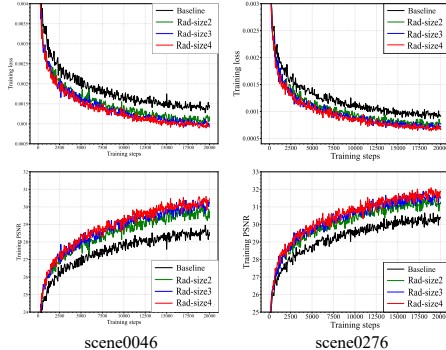

Figure 7: Convergence curve on ScanNet.

training interference from invisible frontside rays. (3) In the Caterpillar scene, the gating module assigns different preferences to foreground/background regions or the different sides of the caterpillar, which are clearly distinguished. The visualization demonstrates that Rad-NeRF learns reasonable ray allocations, matching our intuition. Besides, we observe that in some specific scenes, such as the Truck scene, the gating score visualization indeed shows a significant difference between the edge and the central region, correlating with the aliasing issue. Such observation illustrates that tackling the aliasing issue in some scenarios is another insightful explanation of the Rad-NeRF's effectiveness, which is supplementary to our original motivation targeting scenarios with heavy occlusions.

**Scaling up NeRF with the Rad-NeRF framework is more effective than scaling the MLP width, increasing the feature grid size, or adding more feature grids.** By default, we set the number of sub-NeRFs to 2 in all experiments. As shown in Figure 6, when the number of sub-NeRFs increases, Rad-NeRF consistently obtains average performance gains on the ScanNet dataset while only marginally increasing the number of model parameters. Compared with directly increasing the hidden dimension of MLP decoders or the size of the feature grid, Rad-NeRF has better performance-model size scalability. Furthermore, we observe that the model with four sub-NeRFs converges faster than the one with two sub-NeRFs while achieving better rendering quality with the same training iterations, as Figure 7 shows. The ease of training convergence can be attributed to two aspects. On the one hand, the number of learnable parameters and training complexity increases marginally. On the other hand, our gate module (a 4-layer MLP without sinusoidal position encoding) decouples the training in the ray dimension and reduces training interference.

### 5.3 Comparison with Gaussian Splatting

We additionally compare Rad-NeRF against 3D Gaussian splatting (3DGS) [11] as a non-neural approach that represents the current state of the art with regard to quality and rendering speed. The comparison is conducted on MaskTAT [13] and ScanNet [6] datasets. MaskTAT is an object dataset without point clouds, and ScanNet contains indoor scenes with many less textured regions.

**Rad-NeRF performs better than 3DGS in some cases.** We report results on Table 9, and show qualitative highlights in Figure 8. For the MaskTAT dataset, we initialize 3D GS with random points. Our method performs best over 3D GS and Instant-NGP. For the ScanNet dataset, we initialize 3D GS with the point cloud provided by the dataset. However, there are many less textured regions in

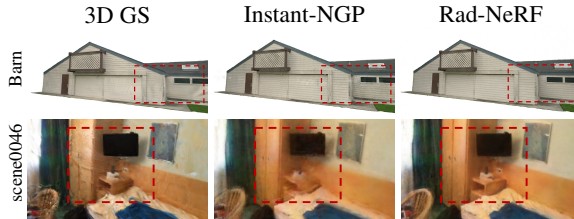

Figure 8: Qualitative comparisons with 3D GS.

| Methods | MaskTAT | ScanNet |
|---|---|---|
|  | PSNR↑ | PSNR↑ |
| 3D GS | 27.363 | 26.781 |
| Instant-NGP | 28.752 | 28.074 |
| Rad-NeRF | **29.774** | **28.870** |

Figure 9: Quantitative results.

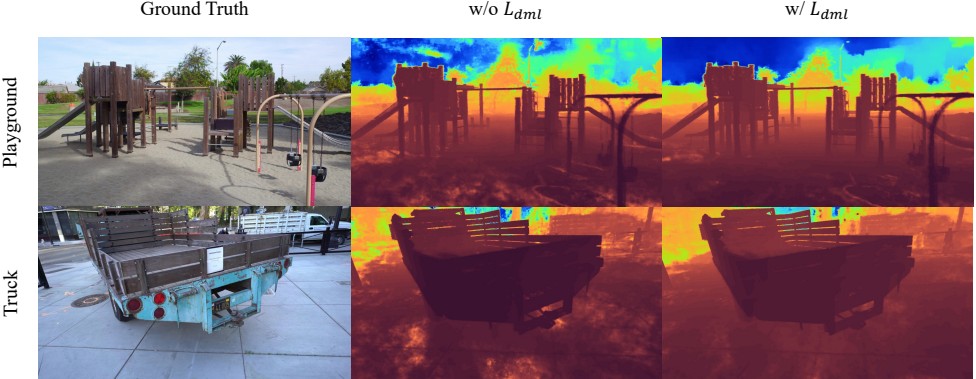

Figure 10: Depth visualization comparison between w/o $L_{dml}$ and w/ $L_{dml}$ on TAT dataset. Zoom in to see the details of sky and ground.

Table 2: Ablation results of gate-guided multi-NeRF fusion and depth-based mutual learning.

| Method | Metric | M60 | Playground | Train | Truck | Avg |
|---|---|---|---|---|---|---|
| Uniform fusion | PSNR↑ | 19.229 | 22.863 | 17.531 | 23.569 | 20.798 |
| | SSIM↑ | 0.633 | 0.694 | 0.596 | 0.746 | 0.667 |
| | LPIPS↓ | 0.431 | 0.414 | 0.451 | 0.345 | 0.411 |
| w/o depth mutual loss | PSNR↑ | 18.912 | 23.399 | 17.371 | 24.665 | 21.087 |
| | SSIM↑ | 0.621 | 0.694 | 0.589 | 0.758 | 0.666 |
| | LPIPS↓ | 0.436 | 0.402 | 0.449 | 0.329 | 0.404 |
| Rad-NeRF | PSNR↑ | 19.051 | 23.901 | 19.369 | 24.509 | **21.708** |
| | SSIM↑ | 0.631 | 0.689 | 0.612 | 0.757 | **0.672** |
| | LPIPS↓ | 0.429 | 0.402 | 0.431 | 0.333 | **0.399** |

indoor scenes that affect the accuracy and density of point clouds. Optical distortion exists in the rendered pictures of 3D GS. In contrast, Rad-NeRF renders more smoothly than all baselines.

**Potential combination of Rad-NeRF with 3DGS.** NeRF is characterized by its neural network-based ray-related predictions, which provide flexibility for cross-scene generalization and enable the application of Rad-NeRF's ray-wise training decoupling approach. In contrast, the plain 3D GS framework parametrizes the scene using a global, non-ray-related representation, making Rad-NeRF inapplicable. However, Rad-NeRF could potentially be applied to generalizable 3D GS frameworks that integrate neural network-based ray-related predictions [4].

## 5.4 Ablation Studies

In this section, we conduct ablation studies on Rad-NeRF using the TAT dataset [13]. The key takeaways from our results are summarized below. Some additional ablation studies and analyses are presented in the Appendix E.

**Importance of the gate-guided multi-NeRF fusion and depth-based mutual learning.** The ablation results of the two key components are shown in Table 2. Uniform fusion simply averages multi-NeRFs' outputs to get final results without a gate module. In this way, sub-NeRFs focus on all the training rays instead of having their own preferences, which can not effectively improve rendering quality. For the depth-based mutual learning method, we observe that it enables a smoother and more reasonable depth prediction, as shown in Figure 10. In addition to improving rendering consistency, it also acts as a geometric regularization to reduce the depth ambiguity and avoid overfitting.

We further provide visualizations of different sub-NeRFs' rendering results in Figure 11, which validates that the proposed depth-based mutual learning scheme will not encourage all sub-NeRFs to converge to the same output. On the one hand, the soft gating module allocates different rays to different sub-NeRFs, making them learn from different views. On the other hand, the depth-based mutual learning scheme only lets sub-NeRFs learn the depth from each other rather than the overall rendered density or RGB distribution.

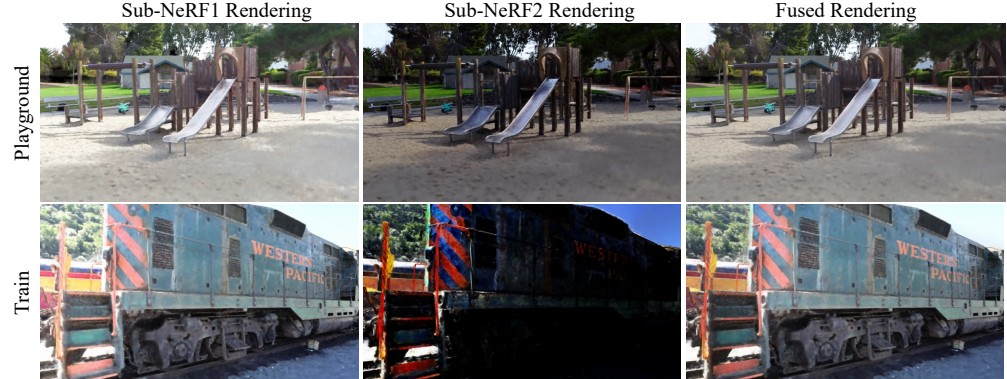

| Sub-NeRF1 Rendering | Sub-NeRF2 Rendering | Fused Rendering |

Figure 11: Independent and fused rendering results of sub-NeRFs on TAT dataset.

Table 3: Ablation results of fusion dimensions.

| Fusion Dimension | PSNR↑ | SSIM↑ | LPIPS↓ |
|---|---|---|---|
| Point-level | 20.796 | 0.661 | 0.413 |
| Ray-level (Ours) | **21.708** | **0.672** | **0.399** |

Table 4: Ablation results of fusion granularity.

| Fusion Granularity | PSNR↑ | SSIM↑ | LPIPS↓ |
|---|---|---|---|
| Image-level | 21.503 | 0.669 | 0.408 |
| Pixel-level (Ours) | **21.708** | **0.672** | **0.399** |

**Importance of the ray-level allocation.** We evaluate the results of different fusion dimensions in Table 3. Compared to fusing multi-NeRFs' outputs in the point dimension, our ray-based method performs better, validating the superiority of the visibility-aware multi-NeRF method.

**Importance of pixel-granularity fusion.** We compare different fusion granularity in Table 4. In image-granularity fusion, all pixels of an image have the same preference for model parameters, which may not be reasonable. An illustrative example is an image capturing both the central object and the background region, such as the *Truck* scene shown in Figure 5. In such a case, the rays hitting these two regions should be assigned different model parameters. In contrast, pixel-granularity fusion provides a more fine-grained understanding of the image and scene.

# 6   Limitations

As the gating module (a 4-layer MLP without sinusoidal position encoding) incorporates smoothness prior implicitly, it exhibits smooth and close scores to the nearest seen view for unseen views. Consequently, the generalization of the gating module relies on sufficient training data, and thus Rad-NeRF does not perform well in the few-shot setting (see Appendix K for more results). On the contrary, the proposed method is suitable for the rendering of complex scenes, which themselves often require sufficient training data.

# 7   Conclusion

This work proposes a ray-decoupled training framework (Rad-NeRF) for neural rendering. To alleviate the issue of the training interference problem in complex scenes, we construct a multi-NeRF framework and decouple the training of NeRFs in the ray dimension. Additionally, we propose a depth-based mutual learning method that improves the multi-NeRF rendering consistency and reduces the depth ambiguity, thereby improving generalization to novel views. Extensive experiments across various datasets validate Rad-NeRF's effectiveness and better performance-parameter scalability.

We prospect for further exploration to fully exploit the potential of Rad-NeRF. Here, we outline several possible directions:(1) As researchers may choose different frameworks based on specific situational requirements, adapting Rad-NeRF to different single-NeRF frameworks including 3D GS (non-neural approach) is a valuable next step. (2) The number of sub-NeRFs can be determined automatically based on scene complexity and training resources. (3) We hope the newly proposed scaling dimension, which increases the number of sub-NeRFs through ray-wise decoupling, will enable modeling of complex scenes in a parameter-efficient manner.

## Acknowledgments

Lidong Guo, Xuefei Ning, Tianchen Zhao, Jincheng Yu, Yu Wang was supported by the National Key R&D Program of China (2023YFB4502200), the National Natural Science Foundation of China (No. 62325405, 62104128, U21B2031, 62204164), Tsinghua EE Xilinx AI Research Fund, Tsinghua-Meituan Joint Institute for Digital Life, and Beijing National Research Center for Information Science and Technology (BNRist).

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

# Appendix

## Table of Contents

# A Comparison with Other Multi-NeRF Methods

The comparison of various multi-NeRF training frameworks is summarized in Table S.1. NeRF++ [37] proposes the sphere inversion transformation to map an infinite space to a bounded sphere firstly, and uses two NeRFs to model the 3D points in foreground and background regions, respectively. It adopts the manual allocation mode as it manually sets the boundary between foreground and background regions. Block-NeRF [26] and Mega-NeRF [27] are two classical ray-based multi-NeRF frameworks, which perform the ray allocation in the image-granularity and pixel-granularity, respectively. The former work trains sub-NeRFs in large-scale road scenes with prior knowledge of the image shooting position distribution on the road, and the latter one trains sub-NeRFs in open drone scenes and allocates the rays by partitioning the intersecting positions between the rays and a horizontal plane. However, they are designed for large road scenes and open drone scenes specifically and need a manually defined allocation rule, which requires prior scene knowledge and cannot be easily adapted to other types of scenes. Switch-NeRF [38] implements a learning-based scene partition scheme motivated by Mixture-of-Experts (MoE) [24]. However, it partitions the scene in the point dimension, which limits the rendering performance in more complex scenes with occlusions. It is also limited to be only used in open drone scenes. F2-NeRF [30] is another point-based multi-NeRF method, which allocates the 3D points to multiple sub-NeRFs in a more elaborate but manual way.

In contrast, our Rad-NeRF performs the allocation and decoupling the training in the ray dimension softly. Acting as a ray-based training framework, Rad-NeRF is "visibility-aware" and achieves higher performance in complex scenes. Moreover, compared to other multi-NeRF methods, Rad-NeRF boosts rendering quality across different types of scenes without the need for prior scene knowledge.

Table S.1: Comparison of multi-NeRF training frameworks. **Headers:** The "Dimension" column indicates the dimension in which the framework divides the training data into multiple sub-NeRFs; The "Allocation mode" column indicates whether the framework divides the training data based on the manually designed rule or in a learnable way; The "Target scene" column indicates the scene that the framework is proposed for specifically.

| Multi-NeRF methods | Dimension | Allocation mode | Target scene |
|---|---|---|---|
| NeRF++ [37] | point-based | manual | no constraint |
| Block-NeRF [26] | ray-based | manual | large road scene |
| Mega-NeRF [27] | ray-based | manual | open drone scene |
| Switch-NeRF [38] | point-based | learnable | open drone scene |
| F2-NeRF [30] | point-based | manual | no constraint |
| Rad-NeRF (ours) | **ray-based** | **learnable** | **no constraint** |

# B Implementation Details

## B.1 Implementation Details of Rad-NeRF

**Architecture Details.** Our Rad-NeRF is built upon Instant-NGP [18] using a third-party PyTorch implementation [3] and costs no more than one hour of training. We follow the original architecture of Instant-NGP with 16 levels of resolution. The hash table length at each resolution is fixed to $2^{19}$. The density and color MLP comprise one and two hidden layers with 64 channels respectively.

**Training Details.** For Instance-NGP and our Rad-NeRF, we train the NeRFs for 20k iterations on a single RTX-3090 GPU. We use Adam optimizer with a batch size of 8192 rays and a learning rate decaying from $1 \times 10^{-2}$ to $3 \times 10^{-4}$. For the weights of the regularization terms in Equation 6, $\lambda_1$ is set to $1 \times 10^{-4}$ on NeRF-360-v2 and Free dataset, and is set to $5 \times 10^{-3}$ on other datasets. We set $\lambda_2$ to $1 \times 10^{-2}$ on all the datasets. By default, the number of sub-NeRFs is set to 2, and it is sufficient to achieve significant rendering quality improvement.

Some previous work has observed that the gate module tends to converge to an imbalanced state, where it always produces large weights for the same few sub-models [23, 28, 38]. Such an imbalance

---

[3]https://github.com/kwea123/ngp_pl

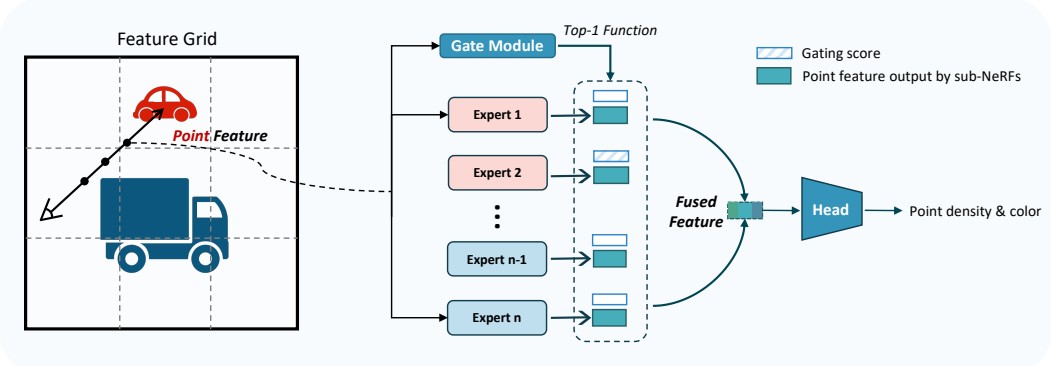

Figure S.1: The overview of Switch-NGP.

problem exists in Rad-NeRF as well. Once the gate module is trapped in a local optimum solution, it will always choose a specific sub-NeRF for rendering and can't effectively decouple the training in the ray dimension.

Following [23, 28], we adopt the regularization on the Coefficient of Variation of the soft gating scores, which encourages a balanced allocation of model parameters for training rays. The CV loss function is given by

$$L_{cv} = \frac{\mathrm{Var}(\overline{G}(\mathcal{R}))}{\left(\sum_{k=1}^{n} \overline{G_k}(\mathcal{R})/n\right)^2},$$ (7)

$$\overline{G_k}(\mathcal{R}) = \sum_{\mathbf{r} \in \mathcal{R}} G_k(\mathbf{r}),$$ (8)

where $\overline{G}(\mathcal{R})$ is the set $\left\{\overline{G_k}(\mathcal{R})\right\}_{k=1}^{n}$. Note that some work also uses the load-balanced loss to encourage multi-models to receive roughly equal numbers of training examples [23, 38]. However, this optimization objective is too strict and unsuitable for our framework.

## B.2 Implementation Details of Switch-NGP

Switch-NeRF [38] constructs a point-based multi-NeRF framework based on MLP-based NeRF structure. Given a 3D point $x$, it first extracts high-level point feature $S(x)$ using a linear layer, which will be sent to the gate module to obtain the gating scores. Then, they apply a Top-1 function on the gating scores to determine which NeRF expert should be activated. The output feature of the chosen expert will be multiplied by the gating score corresponding to the expert and obtain the fused point feature. Finally, the fused point feature is sent to the unified MLPs to predict the density $\sigma$ and color $c$.

As illustrated in Figure S.1, we build an NGP-version of Switch-NeRF, named Switch-NGP. Since NGP contains a feature grid in the form of the hash table, we directly use the feature grid to obtain the high-level point feature $S(x)$ of the point $x$. Switch-NeRF has validated the importance of a unified head, wherein the gating score is multiplied by the high-level features rather than the density or color predictions, which makes the gating and prediction more stable in training. We also perform the multi-NeRF fusion in the point-feature dimension by inserting extra $K$ feature MLPs before the density MLP. Each expert in Switch-NGP corresponds to a tiny feature MLP with two hidden layers and 64 channels.

The training details of Switch-NGP are the same as Rad-NeRF, as described in Section B.1.

## B.3 Implementation Details of Block-NGP

Block-NeRF [26] applies the multi-NeRF method to the street scene, which allocates model parameters in the ray dimension but in the image-level granularity. Specifically, Block-NeRF places one NeRF at each intersection and directly allocates the training images to multi-NeRFs according to the image shooting positions. We implement an NGP-version Block-NeRF, named Block-NGP, which

can be applied to various types of scenes without prior knowledge. After getting all the training images, we first use the clustering algorithm (KMeans) to cluster the image shooting positions, and the number of clusters is set the same as the number of sub-NeRFs. During the training process, each training image is allocated to the corresponding sub-NeRF according to the clustering results, and the training of sub-NeRFs is independent.

## C  Experiments on ScanNet Dataset

We compare Rad-NeRF with other multi-NeRF work on ScanNet dataset [6]. Compared to other outdoor datasets, ScanNet contains more texture-less regions like the floors and the walls, which poses more challenges for neural rendering. We conduct experiments in four complete scenes in ScanNet, namely scene0046, scene0276, scene0515 and scene0673. The quantitative and qualitative results are shown in Table S.2 and Figure S.2 respectively. Our Rad-NeRF outperforms other multi-NeRF methods and renders less blur.

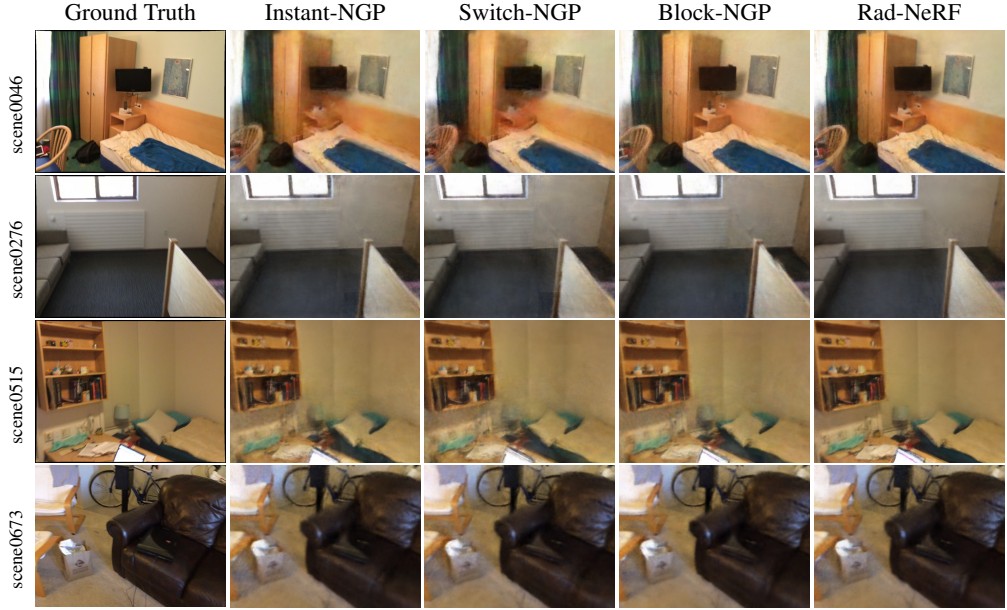

Figure S.2: Qualitative comparisons on ScanNet dataset. Compared to other multi-NeRF methods, Rad-NeRF renders less blur and achieves better recovery of details.

Table S.2: Quantitative results on ScanNet dataset.

| Methods | Metrics | scene0046 | scene0276 | scene0515 | scene0673 | Avg |
|---|---|---|---|---|---|---|
| NGP | PSNR↑ | 28.504 | 29.996 | 28.159 | 25.278 | 27.984 |
|  | SSIM↑ | 0.839 | 0.835 | 0.786 | 0.686 | 0.786 |
|  | LPIPS↓ | 0.413 | 0.421 | 0.448 | 0.472 | 0.438 |
| Switch-NGP | PSNR↑ | 28.135 | 29.614 | 27.814 | 25.140 | 27.676 |
|  | SSIM↑ | 0.834 | 0.831 | 0.779 | 0.684 | 0.782 |
|  | LPIPS↓ | 0.421 | 0.431 | 0.456 | 0.473 | 0.445 |
| Block-NGP | PSNR↑ | 28.728 | 30.214 | 28.332 | 25.444 | 28.180 |
|  | SSIM↑ | 0.842 | 0.840 | 0.789 | 0.688 | 0.790 |
|  | LPIPS↓ | 0.408 | 0.416 | 0.443 | 0.469 | 0.434 |
| Rad-NeRF | PSNR↑ | 29.440 | 30.871 | 29.149 | 25.759 | **28.805** |
|  | SSIM↑ | 0.851 | 0.843 | 0.800 | 0.690 | **0.796** |
|  | LPIPS↓ | 0.396 | 0.405 | 0.427 | 0.469 | **0.424** |

## D  Per-Scene Results

We provide the per-scene quantitative results on the Mask-TAT dataset, TAT dataset, NeRF-360-v2 dataset and Free dataset in Table S.3, Table S.4, Table S.5 and Table S.6 respectively. The results are reported in the metric of PSNR.

Table S.3: Scene breakdown on the Mask-TAT dataset.

| Methods | Ignatius | Truck | Barn | Caterpillar | Family | Avg |
|---------|----------|-------|------|-------------|--------|-----|
| NeRF | 25.43 | 25.36 | 24.05 | 23.75 | 30.29 | 25.78 |
| MipNeRF | 29.037 | 23.19 | 28.481 | 28.016 | 29.009 | 27.547 |
| PlenOctrees | 28.19 | 26.83 | 26.8 | 25.29 | 32.85 | 27.99 |
| DVGO | 28.16 | 27.15 | 27.01 | 26.00 | 33.75 | 28.41 |
| Instant-NGP | 28.431 | 27.562 | 27.611 | 26.065 | 34.092 | 28.752 |
| Switch-NGP | 28.184 | 27.34 | 27.472 | 25.75 | 33.711 | 28.491 |
| Block-NGP | 28.202 | 27.621 | 27.768 | 26.06 | 34.081 | 28.746 |
| Rad-NeRF | 29.806 | 28.163 | 28.701 | 27.445 | 34.756 | 29.774 |

Table S.4: Scene breakdown on the TAT dataset.

| Methods | M60 | Playground | Train | Truck | Avg |
|---------|-----|------------|-------|-------|-----|
| NeRF | 16.86 | 21.55 | 16.64 | 20.85 | 18.975 |
| NeRF++ | 17.964 | 22.914 | 18.194 | 22.603 | 20.419 |
| MipNeRF-360 | 20.091 | 24.27 | 19.741 | 24.144 | 22.062 |
| MipNeRF360$_{short}$ | 18.394 | 22.682 | 17.738 | 21.497 | 20.078 |
| DVGO | 17.292 | 22.62 | 17.783 | 21.306 | 19.750 |
| Instant-NGP | 18.914 | 22.832 | 17.707 | 23.428 | 20.720 |
| Switch-NGP | 18.619 | 22.661 | 17.523 | 23.243 | 20.512 |
| Block-NGP | 18.879 | 22.555 | 18.048 | 23.651 | 20.783 |
| Rad-NeRF | 19.051 | 23.901 | 19.369 | 24.509 | 21.708 |

Table S.5: Scene breakdown on the NeRF-360-v2 dataset.

| Methods | bicycle | bonsai | counter | garden | kitchen | room | stump | Avg |
|---------|---------|--------|---------|--------|---------|------|-------|-----|
| NeRF | 21.818 | 29.028 | 26.980 | 23.640 | 27.164 | 30.097 | 22.934 | 25.952 |
| NeRF++ | 21.426 | 31.670 | 27.717 | 24.801 | 29.468 | 30.621 | 24.770 | 27.210 |
| MipNeRF360 | 22.861 | 32.970 | 29.291 | 26.014 | 31.987 | 32.685 | 25.278 | 28.727 |
| MipNeRF360$_{short}$ | 21.264 | 28.040 | 26.366 | 23.214 | 26.552 | 29.636 | 23.313 | 25.484 |
| DVGO | 21.652 | 27.919 | 26.432 | 23.851 | 26.282 | 31.677 | 20.988 | 25.543 |
| F2-NeRF | 21.311 | 30.036 | 25.873 | 23.694 | 28.935 | 29.421 | 24.251 | 26.217 |
| Instant-NGP | 24.203 | 31.374 | 25.665 | 25.312 | 30.278 | 31.534 | 22.799 | 27.309 |
| Switch-NGP | 23.859 | 30.012 | 24.359 | 25.164 | 29.865 | 31.127 | 21.284 | 26.524 |
| Block-NGP | 24.186 | 31.684 | 25.704 | 25.288 | 30.382 | 31.569 | 23.241 | 27.436 |
| Rad-NeRF | 24.550 | 32.439 | 25.230 | 25.634 | 31.062 | 32.863 | 23.312 | 27.871 |

## E  Additional Ablation Studies

We add additional ablation studies on the TAT dataset to further analyze the mechanism of Rad-NeRF, including structural design, depth-mutual learning, and CV-balanced regularization. The results are shown in Table S.7.

**Gate-guided depth mutual learning.** In Rad-NeRF, we use the gate-guided fused depth as the target depth to regularize sub-NeRFs' geometry and avoid overfitting. By contrast, when we directly use

Table S.6: Scene breakdown on the Free dataset

| Methods | Hydrant | Lab | Pillar | Road | Sky | Stair | Grass | Avg |
|---------|---------|-----|--------|------|-----|-------|-------|-----|
| NeRF | 16.569 | 17.342 | 20.944 | 19.793 | 15.925 | 18.731 | 22.439 | 18.820 |
| NeRF++ | 22.948 | 23.718 | 26.353 | 24.916 | 25.059 | 27.647 | 21.504 | 24.592 |
| MipNeRF360 | 25.03 | 26.57 | 29.22 | 27.07 | 26.99 | 29.79 | 24.39 | 27.008 |
| MipNeRF360$_{short}$ | 23.281 | 24.412 | 26.789 | 24.158 | 25.369 | 27.139 | 21.827 | 24.711 |
| DVGO | 22.315 | 23.123 | 25.345 | 23.242 | 24.736 | 25.844 | 19.794 | 23.485 |
| Instant-NGP | 23.29 | 26.084 | 28.683 | 26.302 | 26.05 | 28.158 | 23.088 | 25.951 |
| F2-NeRF | 24.34 | 25.92 | 28.76 | 26.76 | 26.41 | 29.19 | 22.87 | 26.32 |
| Switch-NGP | 23.197 | 25.901 | 28.080 | 26.155 | 26.034 | 28.097 | 22.819 | 25.755 |
| Block-NGP | 23.663 | 26.682 | 28.103 | 25.989 | 26.283 | 28.395 | 22.988 | 26.015 |
| Rad-NeRF | 24.463 | 25.751 | 28.871 | 26.827 | 27.235 | 28.562 | 23.433 | 26.449 |

the average of the sub-NeRFs' rendering depths as the target depth, which means all sub-NeRFs have equal regularization strength (Equal DML), the rendering quality will be slightly worse. The results highlight the pivotal role of gate-guided depth mutual learning. Using the gated-guided fused depth as the target depth differently penalizes sub-depths based on the gating scores and increases the accuracy of the geometry regularization. We also observe that depth mutual learning has no effect in the case of uniform fusion due to the low accuracy of the averaged depth.

**CV balanced regularization.** As introduced in Section B.1, we adopt the regularization on the Coefficient of Variation of the soft gating scores to prevent the gate module from collapsing onto a specific sub-NeRF while maintaining sub-NeRF's different specialties. Without CV-balanced regularization, the rendering quality degrades significantly. Besides, we apply the CV regularization only for the first half of the training time and find that the performance is comparable to Rad-NeRF, The results prove that such regularization would not interfere with the learning of the gate module.

Table S.7: Additional ablation results.

| Method | Metric | M60 | Playground | Train | Truck | Avg |
|--------|--------|-----|------------|-------|-------|-----|
| Equal DML | PSNR↑ | 18.929 | 23.108 | 19.012 | 24.625 | 21.419 |
| | SSIM↑ | 0.625 | 0.686 | 0.610 | 0.758 | 0.670 |
| | LPIPS↓ | 0.431 | 0.405 | 0.432 | 0.332 | 0.400 |
| Independent feature grids | PSNR↑ | 18.765 | 22.839 | 18.958 | 24.493 | 21.264 |
| | SSIM↑ | 0.625 | 0.697 | 0.614 | 0.762 | 0.675 |
| | LPIPS↓ | 0.431 | 0.405 | 0.417 | 0.325 | 0.395 |
| Uniform fusion w/o DML | PSNR↑ | 19.229 | 22.863 | 17.531 | 23.569 | 20.798 |
| | SSIM↑ | 0.633 | 0.694 | 0.596 | 0.746 | 0.667 |
| | LPIPS↓ | 0.431 | 0.414 | 0.451 | 0.345 | 0.411 |
| Uniform fusion w/ DML | PSNR↑ | 19.005 | 22.766 | 17.532 | 23.513 | 20.704 |
| | SSIM↑ | 0.627 | 0.695 | 0.592 | 0.747 | 0.665 |
| | LPIPS↓ | 0.434 | 0.411 | 0.453 | 0.341 | 0.410 |
| w/o CV loss | PSNR↑ | 18.743 | 22.795 | 17.245 | 23.395 | 20.545 |
| | SSIM↑ | 0.619 | 0.683 | 0.587 | 0.731 | 0.655 |
| | LPIPS↓ | 0.445 | 0.419 | 0.465 | 0.354 | 0.421 |
| Half CV loss | PSNR↑ | 19.114 | 24.003 | 19.462 | 24.518 | 21.774 |
| | SSIM↑ | 0.625 | 0.689 | 0.606 | 0.758 | 0.670 |
| | LPIPS↓ | 0.433 | 0.404 | 0.430 | 0.334 | 0.400 |
| Rad-NeRF | PSNR↑ | 19.051 | 23.901 | 19.369 | 24.509 | 21.708 |
| | SSIM↑ | 0.631 | 0.689 | 0.612 | 0.757 | 0.672 |
| | LPIPS↓ | 0.429 | 0.402 | 0.431 | 0.333 | 0.399 |

**Structural design.** In Rad-NeRF, we adopt a multi-NeRF structure with a shared feature grid and an ensemble of MLP decoders. We further analyze the reason behind the performance improvement and explore the performance of independent feature grids. As Table S.7 shows, the model employing a shared feature grid (Rad-NeRF) outperforms its counterpart with multiple independent feature grids, which highlights the effect of independent MLP decoders rather than feature grids. We attribute this observation and the performance gained by Rad-NeRF to two aspects. (1)Within the hybrid representation, the feature grid is responsible for encoding features of 3D spatial points, while the MLP encoder is designed to encode view information. The crucial design of independent MLP decoders aligns with our visibility-aware motivation, thereby enhancing the view-dependent effect. (2)The training complexity will also increase as the trainable parameters increase. With the limited amount of training data, increasing the number of feature grids leads to poor convergence. By contrast, as different rays may pass through the same region of 3D space, weight sharing for the feature grid helps to facilitate training. Although the number of learnable parameters hardly increases, Rad-NeRF decouples the training in the ray dimension, helping to increase the model's generalization ability.

## F    Discussion of Mega-NGP

Mega-NeRF [27] applies the multi-NeRF method to the drone scenes, allocating model parameters in the ray dimension and the pixel-level granularity. Specifically, it allocates rays by partitioning the intersecting points between rays and scenes. Such a method is suitable for drone scenes, where the top-down perspective allows for the approximation of ray-scene intersections by intersecting with a set horizontal plane. However, in unstructured scenes captured by free trajectories, the intersecting points between rays and scenes cannot be determined before the training is completed, limiting the applicability of Mega-NeRF to such scenes.

Since there is no straightforward implementation to determine the ray intersections before training, we adopt an alternative implementation for NGP-version Mega-NeRF, which employs a clustering algorithm to divide rays directly based on their origins and directions. The clustering process is offline and the same as the one in Block-NGP. During the training process, each training pixel is allocated to one corresponding sub-NeRF according to the clustering results. To ensure a fair comparison, the model structure of Mega-NGP is the same as the one in Rad-NeRF, following the implementation of Block-NGP. We conduct a comprehensive evaluation across all datasets and the experimental results are shown in Table S.8. Mega-NGP yields similar results to Block-NGP, which is less effective than our Rad-NeRF.

Table S.8: Comparison with Mega-NGP and Rad-NeRF

| Method | Metric | TAT | 360v2 | Free Dataset | ScanNet |
|--------|--------|-----|-------|--------------|---------|
| Mega-NGP | PSNR↑ | 20.843 | 27.482 | 25.855 | 28.100 |
| | SSIM↑ | 0.659 | 0.761 | 0.696 | 0.786 |
| | LPIPS↓ | 0.415 | 0.311 | 0.332 | 0.437 |
| Rad-NeRF | PSNR↑ | 21.708 | 27.87 | 26.449 | 28.870 |
| | SSIM↑ | 0.672 | 0.769 | 0.719 | 0.797 |
| | LPIPS↓ | 0.399 | 0.298 | 0.285 | 0.424 |

## G    More Scalability Studies

We provide the per-scene results of scalability studies on the ScanNet dataset in Table S.9 which are reported in the metric of PSNR.

Furthermore, we observe that the model with four sub-NeRFs converges faster than the one with two sub-NeRFs while achieving better rendering quality with the same training iterations, as Figure S.3 shows. The ease of training convergence can be attributed to two aspects. On the one hand, the feature grid is shared among multi-NeRFs, and thus, the number of learnable parameters increases marginally. On the other hand, as the neural network is better at fitting low-frequency information, our gate module (a 4-layer MLP without sinusoidal position encoding) has implicitly incorporated "smoothness prior", leading to closer rays to be more possibly assigned closer gating scores.

Training Loss                                    Training Accuracy

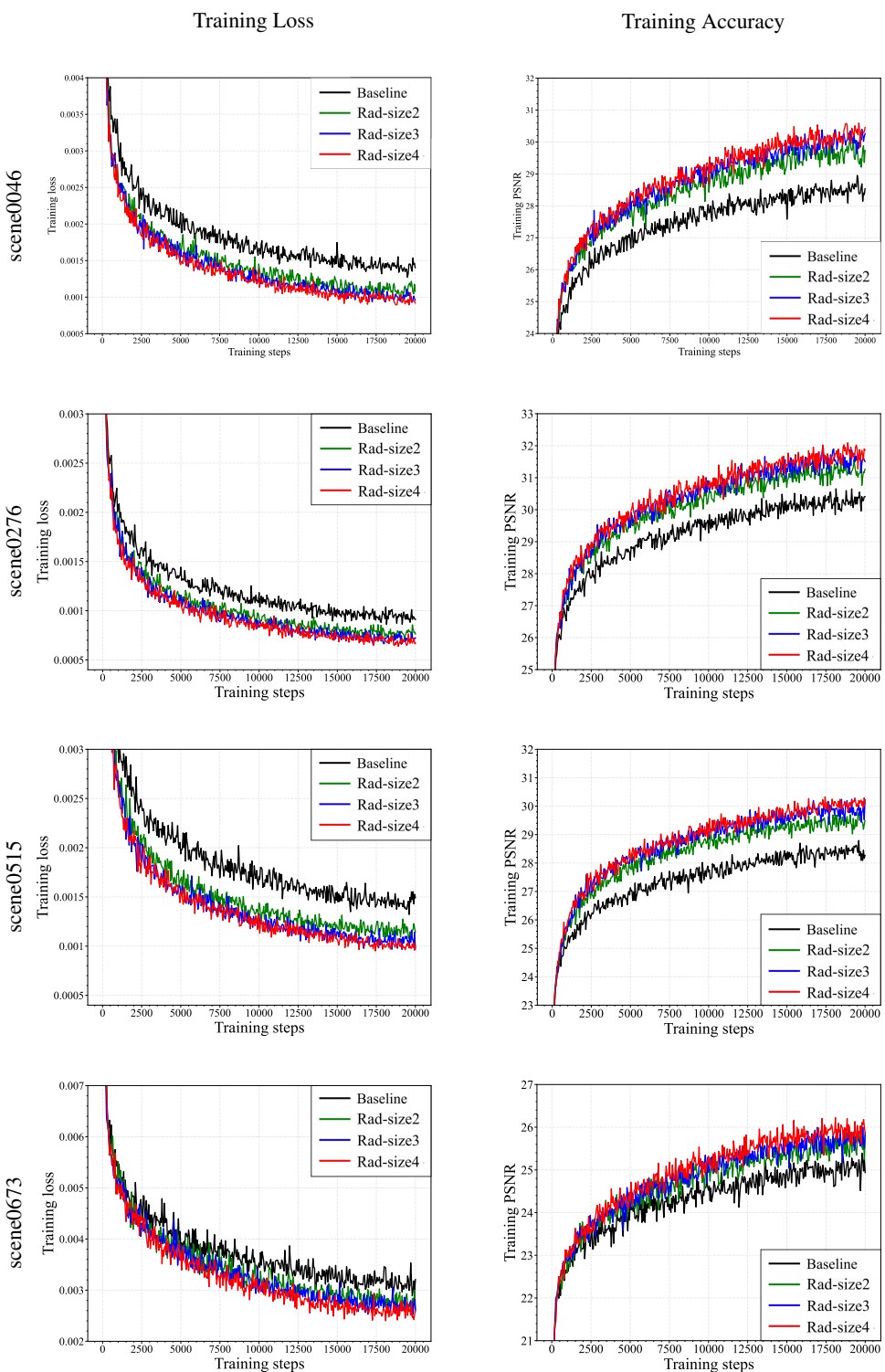

Figure S.3: Convergence curve on the ScanNet dataset.

Table S.9: Scene breakdown of scalability studies on the ScanNet dataset.

| Method | 004600 | 027600 | 051500 | 067304 | Avg |
|---|---|---|---|---|---|
| Instant-NGP | 28.504 | 29.996 | 28.159 | 25.278 | 27.984 |
| Rad-NeRF-size2 | 29.440 | 30.871 | 29.149 | 25.759 | 28.805 |
| Rad-NeRF-size3 | 29.878 | 31.242 | 29.470 | 25.944 | 29.134 |
| Rad-NeRF-size4 | 30.018 | 31.310 | 29.679 | 26.063 | 29.268 |

## H   The Training and Inference Efficiency of Rad-NeRF

We expand the scalability study in the main paper and supplement additional results about training time and inference speed. The comparison results are shown in Figure S.4. Compared to the Instant-NGP baseline, all methods for scaling up NeRF's capacity require longer training time and exhibit lower inference speed, including scaling the MLP width and different multi-NeRF frameworks. Among these methods, Rad-NeRF achieves the best tradeoff between training/inference efficiency and rendering quality. Since we adopt a shared feature grid and multiple independent MLP decoders in the Rad-NeRF framework, a point feature needs to be processed by MLPs in turn, which is the major cause of reduced efficiency. However, as multiple independent MLP decoders can be combined into a single MLP through appropriate parameter initialization and freezing, Rad-NeRF can obtain further efficiency improvements and approach the efficiency of scaling the MLP width.

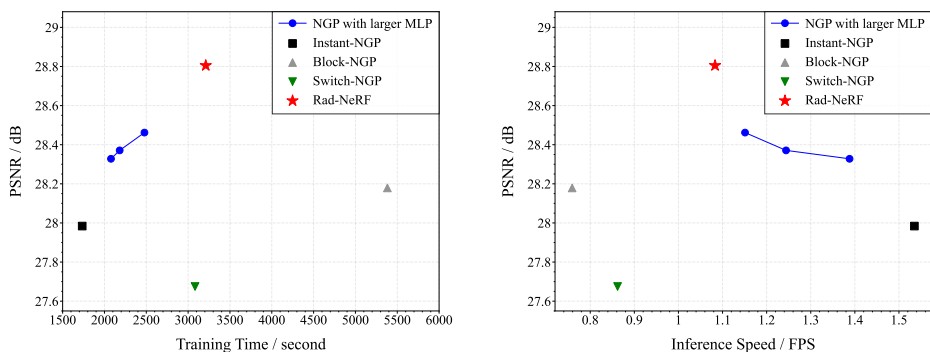

Figure S.4: Scalability study about training/inference efficiency.

## I   Integration of Rad-NeRF and Zip-NeRF

As a multi-NeRF training framework, Rad-NeRF is essentially orthogonal to the structure and training method of single-NeRF. For the benefit of training efficiency and its wide application, we build and validate Rad-NeRF upon the Instant-NGP. Nevertheless, it can also be integrated with other single-NeRF frameworks, such as ZipNeRF [3] (a SOTA single-NeRF framework).

We implement a ZipNeRF version of Rad-NeRF, named Rad-ZipNeRF, and evaluate the performance on the 360v2 dataset. Similar to Rad-NeRF, Rad-ZipNeRF adopts a shared feature grid and multiple MLP decoders. The training settings are kept the same as the original paper, including the training iterations and batch size. As shown in Table S.10, integrated with Rad-NeRF, ZipNeRF can also obtain performance gains, validating Rad-NeRF's effectiveness and potential for integration with different frameworks.

Considering that different frameworks have different characteristics, researchers may choose different frameworks based on specific situational requirements. Adapting Rad-NeRF to different single-NeRF frameworks remains an interesting point to be explored in the future.

We further validate the performance of Rad-NeRF on the Free dataset [34]. As the results show, Rad-NeRF's multi-NeRF training framework boosts ZipNeRF's performance consistently.

Table S.10: Comparison with ZipNeRF and Rad-ZipNeRF on the 360v2 dataset.

| Methods | bicycle | bonsai | counter | garden | kitchen | room | stump | Avg |
|---|---|---|---|---|---|---|---|---|
| ZipNeRF | 21.019 | 33.052 | 25.982 | 24.330 | 32.843 | 34.777 | 25.406 | 28.201 |
| Rad-ZipNeRF | 20.488 | 33.486 | 26.372 | 24.603 | 33.120 | 35.795 | 25.581 | 28.492 |

Table S.11: Comparison with ZipNeRF and Rad-ZipNeRF on the free dataset.

| Method | Hydrant | Lab | Pillar | Road | Sky | Stair | Grass | Avg |
|---|---|---|---|---|---|---|---|---|
| ZipNeRF | 25.402 | 27.827 | 25.132 | 28.882 | 26.993 | 28.187 | 18.461 | 25.841 |
| Rad-ZipNeRF | 25.51 | 28.067 | 25.348 | 29.191 | 27.491 | 28.339 | 18.572 | 26.074 |

## J   Additional Visualizations of Gating Scores

In the visualization results of the main paper, we adopt two sub-NeRFs in all scenes of the TAT dataset. With this setting, the two sub-NeRFs exhibit complementary gating scores for the same view and we omitted the visualization of sub-NeRF2 for brevity in the main paper. We also provide the visualization results of the other sub-NeRF in Figure S.5. As shown in Figure S.5, when rendering in an open scene with fewer occlusions, the gating score exhibits different characteristic and smooth transition according to the ray directions. This visualization further validates our analysis that as a 4-layer MLP without sinusoidal position encoding, the gating module incorporates smoothness prior implicitly. For unseen viewpoints, especially in less-occluded outdoor scenes, the gating module exhibits smooth and close scores to the nearest seen view. The additional visualization results further prove that our original motivation for tackling heavy occlusion by decoupling sub-NeRF training is valid.

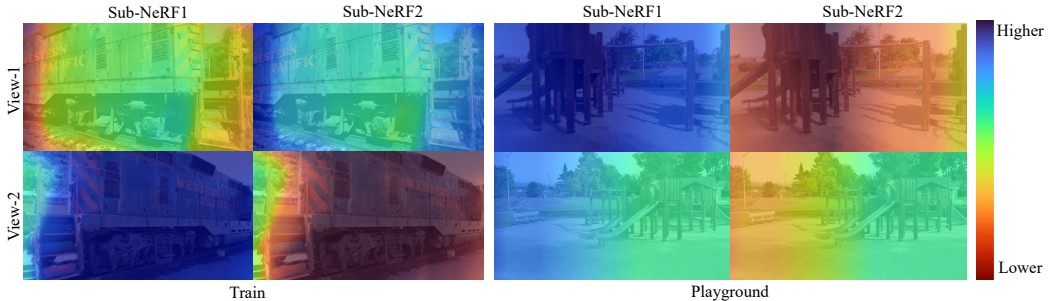

Figure S.5: Additional visualizations of gating scores on two different views on TAT dataset.

## K   Limitation under the Few-shot Setting

Previously, we have included the discussion of the limitation under the few-shot setting. This is because rendering under the few-shot setting presents a greater challenge to both NeRF's and gating module's generalization ability.

We validate Rad-NeRF's performance in the few-shot setting on the LLFF dataset [16]. For 6/9 training views, Rad-NeRF does not exhibit significant benefits or performance improvements compared to Instant-NGP, with all metrics at the same level. This is because insufficient training data affects the training and generalization of the gating module.

However, when rendering with extremely few training data (3 views), Rad-NeRF achieves significantly better rendering quality. We analyze that when training with very few views, the gating module has minimal impact on NeRF's training. Nonetheless, depth-based mutual learning between multiple sub-NeRFs could still exhibit an effective geometric regularization effect, thereby improving rendering performance. This analysis is also validated by the visualization results shown in Figure S.6, compared to the baseline, Rad-NeRF reduces the depth rendering ambiguity and shows better geometry modeling in a 3-view setting.

Table S.12: Rad-NeRF's performance under the few-shot setting

| Method | PSNR↑ | | | SSIM↑ | | | LPIPS↓ | | |
|--------|-------|---|---|-------|---|---|--------|---|---|
| | 3-view | 6-view | 9-view | 3-view | 6-view | 9-view | 3-view | 6-view | 9-view |
| Instant-NGP | 16.107 | 19.594 | 21.105 | 0.419 | 0.592 | 0.663 | 0.541 | 0.394 | 0.353 |
| Rad-NeRF | 16.626 | 19.214 | 20.979 | 0.452 | 0.592 | 0.661 | 0.506 | 0.298 | 0.344 |

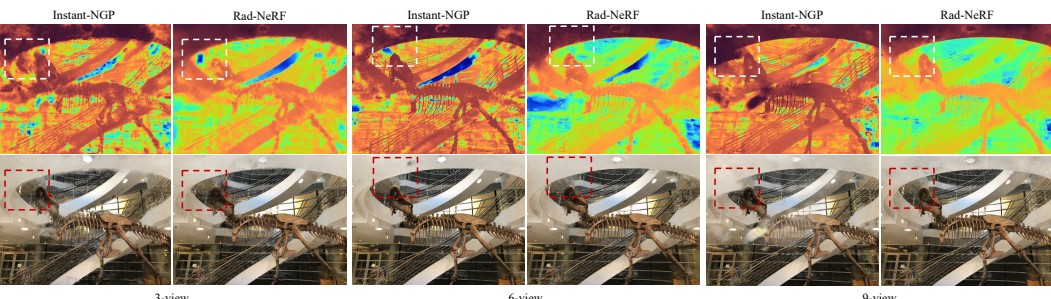

Figure S.6: Qualitative comparisons under three few-shot settings on LLFF dataset.

## L   Comparison of Rad-NeRF with Uncertainty-based Methods

Uncertainty-based methods consider floaters as regions corresponding to high uncertainty and remove them by thresholding the scene according to an uncertainty field during rendering. The spatial uncertainty is computed in roughly a minute on any existing method. For example, RobustNeRF [22] treated pixels with larger losses as those with high uncertainty, avoiding the misleading effect of outlier points by discarding the training of those pixels. However, it is difficult to distinguish outlier points from the high-frequency areas that should be learned. Moreover, Instant-NGP [18] regards the spatial points with too low density as regions with high uncertainty and filters these regions when rendering. Although this method works well, it still cannot completely eliminate floaters in difficult scenes and may remove correct regions. As a post-hoc uncertainty assessment framework, Bayes Rays [9] acts as a post-hoc uncertainty assessment framework, which does not change NeRF's training process, only removing "floater" regions corresponding to high uncertainty. However, this solution is not stable and is generally used as an auxiliary solution to improve NeRF's rendering quality.

Different from uncertainty-based methods, the proposed Rad-NeRF improves rendering quality by tackling the training interference issue. The depth-based mutual learning method also acts as a geometric regularization to reduce rendering defects. Importantly, Rad-NeRF is essentially orthogonal to these post-training uncertainty removal-based methods and can be integrated with Bayes Rays to obtain further performance improvement.

