# OpenReview forum: "Rad-NeRF: Ray-decoupled Training of Neural Radiance Field"
_NeurIPS.cc/2024/Conference — NeurIPS 2024 poster_

### Official Review · Reviewer_5G52 · 2024-06-22

**Soundness:** 3
**Presentation:** 3
**Contribution:** 2
**Rating:** 6
**Confidence:** 3

**Summary:**

This paper claims that training with those rays with invisible 3D points (occlusions in complex scenes) that do not contain valid information about the point might interfere with the NeRF training.

Based on this intuition, this paper proposes Rad-NeRF to decouple the training process of NeRF in the ray dimension softly, construct an ensemble of sub-NeRFs and train a soft gate module to assign the gating scores to these sub-NeRFs based on specific rays, where the gate module is jointly optimized with the sub-NeRF ensemble to learn the preference of sub-NeRFs for different rays.

This paper also introduces depth-based mutual learning to enhance the rendering consistency among multiple sub-NeRFs and mitigate the depth ambiguity.

Experiments on five datasets demonstrate that Rad-NeRF can enhance the rendering performance across a wide range of scene types compared with existing Instant-NGP-based methods.

**Strengths:**

This paper proposes an Instant-NGP-based ray-decoupled training framework to mitigate the training interference caused by invisible rays by ensemble sub-NeRFs via a jointly optimized gate module.

This paper also proposes a depth-based mutual learning method to ensure the rendering consistency among multiple sub-NeRFs, which serves as a geometric regularization, alleviating the depth ambiguity and avoiding overfitting.

This paper conducts extensive experiments. The results show that Rad-NeRF consistently improves rendering quality effectively and obtains better scalability than Instant-NGP-based baselines.

**Weaknesses:**

My main concern is the novelty. Although this work aims for different tasks, it is quite like Switch-NeRF.

**Questions:**

The improved 3DGS nowadays is quite good and does not have the ray sampling problem, so I think this work may have limited influence and help a little for our community. I would like to see the rebuttal and other reviews.

**Limitations:**

The authors adequately addressed the limitations.

---

> ### Author Rebuttal · Authors · 2024-08-07
>
> **R4-Q1: My main concern is the novelty. Although this work aims for different tasks, it is quite like Switch-NeRF.**
>
> Thanks for this question. In our opinion, the most fundamental difference between Switch-NeRF and our Rad-NeRF is that Rad-NeRF is a ray-based multi-NeRF framework (the first ray-based framework as far as we know), while Switch-NeRF is a point-based multi-NeRF framework. And these two types of designs target different objectives. Specifically, Switch-NeRF decomposes 3D points to different sub-NeRFs to **model larger scenes**, and Rad-NeRF decomposes rays to tackle **the training interference from invisible rays**, which is especially of concern in **complex scenes**.
>
> Figure 5 and our newly added Figure R.1 show that scaling the number of sub-NeRFs by decoupling rays to different sub-NeRFs offers better performance-parameter scalability than scaling by decomposing spatial positions or scaling the MLP or spatial grid. We believe that in future applications of scaling NN-based 3D modeling methods for complex scenes, our newly proposed dimension (scaling the number of sub-NeRFs by decoupling rays) is a worth-considering dimension in the compound scaling strategy.
> Besides the major novelty in designing the ray-based multi-NeRF framework, we would also like to mention our introduction of mutual depth supervision between sub-NeRFs. This technique is simple but effective and can be easily adopted in other multi-NeRF frameworks too, providing a new way of unsupervised geometric regularization without the need for manually defined rules.
>
> **R4-Q2: The improved 3DGS nowadays is quite good and does not have the ray sampling problem, so I think this work may have limited influence and help a little for our community.**
>
> Thanks for this very worth-discussing topic! We would like to share our thoughts on this topic and welcome any further discussions. Indeed, 3D GS has surpassed NeRF's rendering quality and speed on quite some tasks. Nevertheless, we still think the NeRF framework featured by the parametric neural network representation together with the volume rendering method still has its merits. Maybe one of its most notable strengths lies in its adaptability for generalizable modeling, i.e., in generalizable modeling tasks where we need to leverage the knowledge of existing scenarios to better and more efficiently model some unseen scenarios. In this case, it's very easy to extend the parametric neural network to take in other forms of input other than the position and ray direction. For example, the neural network can take an image, a depth image or even other modalities of information as input to model the current scenario. We think this flexibility is beneficial for pushing towards general 3D modeling or generation.
>
> Besides, there are some scenarios that 3DGS methods cannot handle well, due to the inaccurate and sparsity of point clouds (such as less-textured regions) [1], and the limitation to reason about view-dependent effects (such as reflection and refraction effects)[2]. In these scenarios, NeRF-based methods can offer better visual quality.
>
> [1] PointNeRF++: A multi-scale, point-based Neural Radiance Field
>
> [2] HybridNeRF: Efficient Neural Rendering via Adaptive Volumetric Surfaces

---

> > ### Comment · Reviewer_5G52 · 2024-08-11
> >
> > Thanks for the rebuttal. It somehow addressed my concerns. However, I don't agree with the discussion with respect to generalizable modelling, as there are many 3DGS-based works (such as pixelSplat, MVSplat and LaRa) that show impressive generalizability.

---

> > > ### Author Response · Authors · 2024-08-12
> > > **Thanks for the feedback and further discussion from the authors**
> > >
> > > Dear reviewer,
> > >
> > > Thanks for the feedback and the follow-up discussion! Let us continue the discussion.
> > >
> > > ---
> > >
> > > We find our previous answer is not so clear in conveying our points. We were comparing the NeRF framework that is featured by (1) volume rendering; (2) NN parametrized, and the NN outputs **ray-related** volumetric information,  to the plain 3DGS framework that is featured by (1) rasterization rendering; (2) Globally-parametrized **non-ray-related** information. In one word, we're not suggesting that 3DGS cannot be effectively extended to the generalizable setting, but stating that **the NN-based ray-related prediction, a feature of NeRF, offers flexibility for generalization**. **This feature can surely be integrated with the plain 3DGS framework**, as done in the works mentioned by the reviewer. For example, pixelSplat adopts a neural network to predict **per-ray(pixel)** Gaussian primitive parameters.
> > >
> > > Importantly, **the involvement of a neural network in predicting ray-related information allows for the application of the ray-wise training decoupling concept from Rad-NeRF**. This approach can effectively address training interference issues arising from inaccurate spatial sampling in complex and heavily occluded scenes.
> > >
> > > That is to say, although the ray-wise decoupled training method of Rad-NeRF may not be applicable for the plain 3DGS framework with global parametrization, **it could potentially be applied to generalizable 3D GS frameworks that incorporate NN-based ray-related prediction**. This can help mitigate its training interference (Note that regardless of whether rasterization or volume rendering is used, the inaccurate sampling issue will exist, and training interference will arise from the inaccurate sampling if using NN-based ray-related prediction). For instance, a possible way to apply the ray-wise decoupling idea on generalizable-3DGS, taking pixelSplat as the example, is constructing multiple NN-based encoders and fusing the per-ray predictions. Surely, there are still things to design to make this work. Nevertheless, considering that the scope of our paper doesn't include the generalizable setting, we leave it to future work and will add a future work discussion.
> > >
> > > ----
> > >
> > > Thanks again for this valuable discussion. It has not only helped improve our paper but also deepened our understanding of the topic. We're open to further discussion.

---

> > > > ### Comment · Reviewer_5G52 · 2024-08-12
> > > >
> > > > Thanks for the reply. It addressed my concerns. Based on other reviews and the rebuttal, especially the discussion of novelty and the discussion against 3D GS, which should be added to the camera ready, I raised my rating to Weak Accept.

---

> > > > > ### Author Response · Authors · 2024-08-12
> > > > >
> > > > > We're very happy that our response has addressed your concerns! Thanks again for all the helpful suggestions and discussions We will add the discussions to the revision.

---

### Official Review · Reviewer_PUvY · 2024-07-08

**Soundness:** 3
**Presentation:** 3
**Contribution:** 2
**Rating:** 6
**Confidence:** 5

**Summary:**

This paper proposes an innovative approach to enhance NeRF performance. The key observation is that due to occlusion, some objects may appear in one ray but not in another. While NeRF uses transmittance to mitigate this issue, the paper argues that this may not be sufficient. To address this, the authors propose using multiple MLPs to decode the same feature and jointly learn a gating function to fuse all the information. The results demonstrate that this design successfully improves performance, albeit by a small margin.

**Strengths:**

+ The proposed idea is innovative and interesting.
+ The approach results in a performance improvement.

**Weaknesses:**

I have several concerns regarding the motivation, methodology, and validation of the proposed approach.

## Motivation:

+ The motivation is not very clear. Specifically, Figure 1 and lines 29-40 argue that poor performance is due to occlusion, leading the authors to propose using multiple MLPs. However, I find Figure 1, especially Figure 1(c), inaccurate. I would expect a CDF instead of a PDF, as NeRF often compensates RGB appearance with poor geometry. Thus, (c) might still have the correct RGB, i.e., its CDF is accurate.
+ Moreover, Figure 7 does not support the occlusion hypothesis. I suspect the issue might be related to aliasing, where the pixel footprint differs between the center and boundary regions. This can also explain why MipNeRF360 outperforms the proposed method, and the comparison between ZipNeRF and RadZipNeRF in the supplementary material is very close.

## Methodology:

The approach resembles MOE or a multi-head transformer. However, I have two questions:

+ First, the proposed method applies depth consistency. Isn't this encouraging all MLPs to converge to the same output? This might explain why rad2, rad3, and rad4 in Figure 6 are very similar.

+ Second, regarding the soft gating, the current input is the ray origin and direction. It might be more effective to include the 3D position, as splitting MLPs based on different spatial regions could be more reasonable. Additionally, the analysis of the gating function (Figure 7) lacks depth, and more details would be appreciated.

## Validation:

+ While the rad approach does improve performance, the improvement is relatively marginal. Furthermore, MipNeRF360 outperforms the proposed method, indicating that the proposed approach is not strong enough.

+ In Table 1, why are switch NeRF and block NeRF not better than instant NGP? After ray participation, they should outperform the vanilla version.

**Questions:**

See above

---

> ### Author Rebuttal · Authors · 2024-08-07
>
> We thank the reviewer for the constructive feedback and thoughtful comments. We address the detailed questions and comments below.
>
> **R3-Q1: Clarification of Figure.1(c) and motivation**
>
> As the reviewer said, NeRF often compensates for RGB appearance with poor geometry. Figure 1(c) might have the correct RGB with inaccurate geometry. This can be seen in the inaccurate sampling probability presented in Figure 1(c), i.e. the PDF distribution of wi in Equation (1). With Figure 1(c), we aim to illustrate that as the NeRF model is apt to learn a poor geometry, undesirable inter-ray training interference can occur during training. Specifically, when rendering ray-3, it is possible to sample the 3D points near the distant object due to inaccurate sampling. Therefore, the modeling of the distant object will be affected by ray-3 color supervision, resulting in the issue of training interference, which is the main motivation of our work.
>
> We're not very sure how the reviewer thinks Figure 1(c) should be changed. Could you please provide more details on this comment? We are eager to understand your perspective better and address any concerns you might have.
>
> **R3-Q2: Figure 7 does not support the occlusion hypothesis. I suspect the issue might be related to aliasing, where the pixel footprint differs between the center and boundary regions. This can also explain why MipNeRF360 outperforms the proposed method, and the comparison between ZipNeRF and RadZipNeRF in the supplementary material is very close.**
>
> Thank the reviewer for the insightful analysis and careful observation. In the Truck scene, the gating score visualization indeed shows a significant difference between the edge and the central region, correlating with the aliasing issue. We agree that tackling the aliasing issue in some scenarios is an insightful explanation of the Rad-NeRF's effectiveness, which is supplementary to our original motivation targeting scenarios with heavy occlusions. We'll add this discussion to the revision.
>
> That being said, we find that our original motivation for tackling heavy occlusion by decoupling sub-NeRF training is valid. We further provide more gating score visualization results. As shown in Figure R.3 (Global Response), the gating module assigns different preferences to foreground/background regions or the different sides of the caterpillar. Additionally, Figure R.4 shows that when rendering a less-occluded open scene, the gating score exhibits different characteristics and smooth transitions according to the ray directions.
>
> **R3-Q3: Will the proposed depth regularization encourage all MLPs to converge to the same output?**
>
> The proposed depth-based mutual learning scheme does not encourage all sub-NeRFs to converge to the same output. We provide visualizations of different sub-NeRFs' rendering results in Figure R.5 (Global Response), which validates our analysis and conclusion.
>
> On the one hand, the soft gating module allocates different rays to different sub-NeRFs, making them learn from different views. On the other hand, the depth-based mutual learning scheme only lets sub-NeRFs learn the *depth* from each other rather than *the overall rendered density or RGB distribution*.
>
> As for the training curves in Figure.6, Rad-NeRF with more sub-NeRFs converges faster while achieving better rendering quality with the same training iterations. However, as mentioned in the reply to the reviewer tZHj, the rendering performance of Rad-NeRF will gradually saturate as the number of sub-NeRFs increases. Therefore, rad-4 and rad-3 are relatively similar.
>
> **R3-Q4.1: Regarding the soft gating, the current input is the ray origin and direction. It might be more effective to include the 3D position, as splitting MLPs based on different spatial regions could be more reasonable.**
>
> Thanks for this worth-discussing question. Splitting MLPs based on spatial regions (the point-based multi-NeRF framework discussed in our paper) and splitting MLPs based on rays (as far as we know, RadNeRF is the first attempt) are two orthogonal methods targeting **different objectives**.
>
> Specifically, our ray-based multi-NeRF framework tackles **the training interference from invisible rays**, which is especially of concern in **complex scenes**. Whereas point-based multi-NeRF frameworks that split MLPs based on 3D spatial regions mainly aim to **model large scenes by decomposition**, which cannot effectively address the training interference issue between invisible rays. For instance, with two sub-NeRFs assigned to 3D points around two objects as shown in Figure 1, ray-3 information pertaining to the central object is still used to train the sub-NeRF associated with the distant object due to inaccurate sampling, potentially resulting in training interference. In contrast, constructing a ray-based gating module-based multi-NeRF framework is a more effective scheme for the training interference issue.
>
> We also conducted a comparison with state-of-the-art point-based multi-NeRF frameworks in Table 1 and conducted the point-based v.s. ray-based ablation study in Table 3. The results show that our ray-based design leads to consistent improvements.
>
> *Due to the character limitation, we'll reply to the remaining questions in the comments.*

---

> ### Author Response · Authors · 2024-08-07
> **Response to the remaining questions (Q4.2, Q5 and Q6)**
>
> **R3-Q4.2: The analysis of the gating function (Figure 7) lacks depth, and more details would be appreciated.**
>
> Thanks for the question! We're not sure does the reviewer means to "add the depth information into Figure 7"? If so, considering that we can directly judge the depth information of the scene by observing the rendered RGB image and thus analyze its correlation with the gating score, we did not add the depth visualization results in Figure 7. As shown in the visualization of Global Response, the gating scores are not always directly related to the scene depth, although some score visualizations show consistency with pixel depths in some views.
>
> If the reviewer is asking for more analyses: We further analyze the additional gating score visualization. As shown in Figure R.3, the gating module can assign different preferences to different sides of the central object. Besides, when the foreground and background of the scene can be clearly distinguished, the preferences of the sub-NeRFs in the fore/background area are also clear (the visualization of view-1 in Figure R.3). However, it is difficult to further summarize the distribution law of the gating scores. Unlike point-based partitioning, the ray-based gating module encodes both the ray origins and directions, incurring higher partition complexity.
>
> **R3-Q5: Consideration of performance improvement**
>
> Thanks for this question. We'd like to share our thoughts and welcome any further discussions.
>
> Firstly, we regard RadNeRF as an easily **integratable** and **workable** strategy to try out in future NeRF applications, as it doesn't require hyperparameter tuning to work well across different types of scenarios, all achieving some improvements (1.02 on mask-TAT, 0.98 on TAT, 0.57 on 360v2, 0.49 on free dataset and 0.82 on ScanNet dataset). For example, in complex scenarios such as the 360v2 dataset, our method can be plugged in and improve the state-of-the-art. Specifically, ZipNeRF obtains 0.74 PSNR gain compared to MipNeRF360. By integrating our method into the ZipNeRF framework, we easily achieve an additional 0.3 PSNR improvement.
>
> Additionally, we compare Rad-NeRF, the first ray-based multi-NeRF framework to our knowledge, with other multi-NeRF training frameworks. The results demonstrate that the ray-based design has distinct advantages over existing point-based designs.
>
> Finally, Figure 5 and our newly added Figure R.1 show that scaling the number of sub-NeRFs by decoupling rays to different sub-NeRFs offers better performance-parameter scalability than scaling by decomposing spatial positions or scaling the MLP or spatial grid. We believe that in future applications of scaling NN-based 3D modeling methods for complex scenes, our newly proposed dimension (scaling the number of sub-NeRFs by decoupling rays) is a worth-considering dimension in the compound scaling strategy.
>
> **R3-Q6: The effect of Switch-NGP and Block-NGP**
>
> Switch-NGP partitions the scenes in the point dimension, which is fundamentally different from our ray-based multi-NeRF framework. In complex scenes with many occlusions and arbitrary shooting trajectories, Switch-NeRF does not consider the different visibility of a target region to different views and can not tackle the training interference effectively, as noted in the reply to Q1. In addition, Switch-NGP does not handle consistency between multiple sub-NeRFs well due to the lack of overlapping regions and proper regularization between sub-NeRFs.
>
> Block-NGP directly allocates the training images to multiple sub-NeRFs according to the image shooting positions. Judging the relationship between images based solely on shooting positions is insufficient for complex scenes, so it will lead to inaccurate partitioning results and unsatisfactory rendering quality.

---

> > ### Comment · Reviewer_PUvY · 2024-08-10
> >
> > I thank the authors for the detailed responses.  Many of my concerns are addressed, which I appreciate a lot.  Below are some new comments.
> >
> > 1) motivation of radnerf: I am satisfied that the new figures demonstrate the evidence of addressing occlusion, e.g. rebuttal fig.R.3.  I would encourage the authors to add discussion about both occlusion and aliasing in the motivation.
> >
> > 2) As for Fig,1 c in the main paper,  I think drawing both pdf and cdf curves, e.g., fig.2 in nues paper(https://arxiv.org/pdf/2106.10689) is better.  I think the current version does not look very great.
> >
> > 3) for the question:
> > > R3-Q4.1: Regarding the soft gating, the current input is the ray origin and direction. It might be more effective to include the 3D position, as splitting MLPs based on different spatial regions could be more reasonable.
> >
> > I have a follow-up question, would it increase the overfitting problem if just using ray origins and directions?  For example, given the tank example, what happens if you view it from the top?   However,  I would accept any results about that.  I think it is great to have it in the paper, even just for failure cases.
> >
> > 4) for the question:
> > > the analysis of the gating function (Figure 7) lacks depth, and more details would be appreciated.
> > Here "depth" means more analysis.  But I am satisfied with the new results in the rebuttal.
> >
> > 5) as for question:
> > > R3-Q5: Consideration of performance improvement
> > I am not very sure about this part:  "By integrating our method into the ZipNeRF framework, we easily achieve an additional 0.3 PSNR improvement."  Is there a new experiment you have did that zipner+radnerf will have further gain?  If so, it should be added in the revision. If not, I think it is best to have it to demonstrate radnerf.
> >
> > Overall my biggest remaining question is about zipnerf+radnerf.  If the authors show it can help zipnerf, this paper should be accepted.

---

> > > ### Author Response · Authors · 2024-08-10
> > > **Reply to the comments of Reviewer PUvY**
> > >
> > > Thanks for your feedback and we're glad our rebuttal has addressed most of the concerns!
> > >
> > > **C1: Motivation of radnerf: I am satisfied that the new figures demonstrate the evidence of addressing occlusion, e.g. rebuttal fig.R.3. I would encourage the authors to add discussion about both occlusion and aliasing in motivation.**
> > >
> > > Thanks again for this insightful analysis. We'll add the discussion about aliasing to the revision in the motivation part.
> > >
> > > **C2: As for Fig,1 c in the main paper, I think drawing both pdf and cdf curves, e.g., fig.2 in nues paper(https://arxiv.org/pdf/2106.10689) is better. I think the current version does not look very great.**
> > >
> > > Thanks for the suggestion. We will add the CDF curve to the revision.
> > >
> > > **C3: Would it increase the overfitting problem if just using ray origins and directions? For example, given the tank example, what happens if you view it from the top? However, I would accept any results about that. I think it is great to have it in paper, even just for failure cases.**
> > >
> > > Thanks for this great question. The ray-based gating module in Rad-NeRF does face the overfitting issue, especially under the few-shot setting. We also included the discussion of this limitation in the paper. For Rad-NeRF to achieve its full performance improvement, the soft-gating module needs to see enough input ray origins and directions to generalize well to other views. In the rebuttal to Reviewer nJqP, we validate Rad-NeRF's performance in the few-shot setting. The results do show that Rad-NeRF does not exhibit significant performance improvements compared to Instant-NGP (with all metrics at the same level) under the few-shot setting. Under such circumstances, the gating module has minimal impact on NeRF's training, but our depth-based regularization can still exhibit a positive effect, as shown in Figure R.2 (Global Response).
> > >
> > > We further visualize the results of the gating module with limited generalization ability and find that the gating scores are not reasonably allocated, confirming the above limitation. We will supplement all the related results and visualization to the Appendix revision.
> > >
> > > **C4: For the question: the analysis of the gating function (Figure 7) lacks depth, and more details would be appreciated. Here "depth" means more analysis. But I am satisfied with the new results in the rebuttal.**
> > >
> > > Thank the reviewer for the clarification. We will add the additional gating score visualizations and analysis to the Appendix revision.
> > >
> > > **C5: Consideration of performance improvement I am not very sure about this part: "By integrating our method into the ZipNeRF framework, we easily achieve an additional 0.3 PSNR improvement." Is there a new experiment you have did that zipner+radnerf will have further gain? If so, it should be added to the revision. If not, I think it is best to have it to demonstrate radnerf.**
> > >
> > > Previously, we list the results of Rad-ZipNeRF on the 360v2 dataset in Appendix H of our paper. Integrated with Rad-NeRF, ZipNeRF can also obtain an additional 0.3 PSNR improvement. During the rebuttal, we further validate the performance of Rad-NeRF on the Free dataset. As the results show, Rad-NeRF's multi-NeRF training framework boosts ZipNeRF's performance consistently. We will add these additional quantitative results in the Appendix revision.
> > >
> > > |    Method   | Metric | Hydrant | Lab    | Pillar | Road   | Sky    | Stair  | Grass  | Avg     |
> > > |-------------|--------|---------|--------|--------|--------|--------|--------|--------|---------|
> > > | ZipNeRF     | PSNR   | 25.402  | 27.827 | 25.132 | 28.882 | 26.993 | 28.187 | 18.461 | 25.841  |
> > > |             | SSIM   | 0.813   | 0.897  | 0.743  | 0.879  | 0.884  | 0.851  | 0.311  | 0.768   |
> > > | Rad-ZipNeRF | PSNR   | 25.510   | 28.067 | 25.348 | 29.191 | 27.491 | 28.339 | 18.572 | **26.074**  |
> > > |             | SSIM   | 0.812   | 0.903  | 0.748  | 0.866  | 0.891  | 0.852  | 0.317  | **0.770**   |

---

> > > > ### Comment · Reviewer_PUvY · 2024-08-11
> > > >
> > > > Thanks for the detailed responses!    I am very happy that my concerns are resolved and I would like to raise the score as weak accept.

---

> > > > > ### Author Response · Authors · 2024-08-11
> > > > >
> > > > > Thanks again for all the thoughtful suggestions, which are very helpful in improving our work. We will incorporate them into the revision.

---

### Official Review · Reviewer_tZHj · 2024-07-12

**Soundness:** 3
**Presentation:** 3
**Contribution:** 3
**Rating:** 5
**Confidence:** 5

**Summary:**

Traditional NeRF models face challenges in rendering complex scenes due to interference from occluded rays, which leads to inaccurate training data. To address this, the authors propose Rad-NeRF, a framework that decouples the training process in the ray dimension by using multiple sub-NeRFs, each trained with specific rays that contain valid information for the points they observe.
Rad-NeRF employs a soft gate module that assigns gating scores to different sub-NeRFs based on the rays, allowing the model to learn preferences for different rays. Additionally, the framework introduces depth-based mutual learning to enhance consistency and mitigate depth ambiguity among the sub-NeRFs. Essentially, my understanding is that this work builds on Instant NGP by adding multiple sub-MLPs, with the gate module controlling the degree of participation of each MLP during rendering. Experiments demonstrate that Rad-NeRF outperforms existing single and multi-NeRF methods, improving rendering performance with minimal additional parameters.

**Strengths:**

1.	The paper is well-organized, with a clear abstract, introduction, methodology, experiments, and results sections.
2.	The innovative approach of decoupling training in the ray dimension helps mitigate training interference caused by invisible rays. This leads to more accurate and consistent rendering results.
3.	The introduction of a soft gate module that learns the preference of sub-NeRFs for different rays is a novel and effective way to allocate rays dynamically. This approach eliminates the need for manually defined allocation rules, making the method more adaptable and generalizable.
4.	Rad-NeRF enhances rendering consistency among multiple sub-NeRFs through depth-based mutual learning. This technique also helps reduce depth ambiguity and improves geometric modeling accuracy.
5.	Rad-NeRF can be integrated with various single-NeRF frameworks, further improving their performance. For instance, integrating with ZipNeRF shows potential for even better rendering outcomes, validating the flexibility and compatibility of Rad-NeRF with different neural rendering approaches.

**Weaknesses:**

1.	What is the structure of the grid in the network architecture? Is it a multi-resolution grid? Are the features on the grid also learnable?
2.	What are the training and inference times for this method? It would be beneficial if the authors provided a comparison with other existing methods in the paper.
3.	In the case of multiple networks, if a sub-NeRF receives very low soft gating scores, does it cease to update, or does it still contribute to the computation?
4.	Are there any automated methods or heuristics for selecting the optimal number of sub-NeRFs based on scene complexity? What is the relationship between the number of sub-models and the overall performance? Detailed tuning and experimentation are required to ensure optimal performance across different datasets and scenes.
5.	Could you provide a visualization of the gating scores for different sub-NeRFs for the same viewpoint? And I am curious about the distribution of gating scores in scenarios without occlusions.

**Questions:**

See Weakness.

**Limitations:**

See Weakness.

---

> ### Author Rebuttal · Authors · 2024-08-07
>
> We thank the reviewer for the constructive feedback and thoughtful comments. We address the detailed questions and comments below.
>
> **R2-Q1: Clarification of the network structure**
>
> Within Rad-NeRF, we adopt a multi-resolution learnable feature grid shared among all sub-NeRFs. Given a 3D point coordinate, we find the surrounding voxels at different resolution levels, index the hash table according to each vertex position, and obtain the final point feature through linear interpolation. During training, loss gradients are backpropagated through the multiple independent MLP decoders, the gating module, and then accumulated in the looked-up learnable feature vectors. We will demonstrate this detail clearly in the revised version.
>
> **R2-Q2: Comparison of training and inference time**
>
> Following the reviewer's advice, we expand the scalability study in the main paper by supplementing additional results about training time and inference speed. The comparison results are shown in Figure R.1 (Global Response). Compared to the Instant-NGP baseline, all methods for scaling up NeRF's capacity require longer training time and exhibit lower inference speed, including scaling the MLP width and different multi-NeRF frameworks. Among these methods, Rad-NeRF achieves the best tradeoff between training/inference efficiency and rendering quality. Since we adopt a shared feature grid and multiple independent MLP decoders in the Rad-NeRF framework, a point feature needs to be processed by MLPs in turn, which is the major cause of reduced efficiency. However, as multiple independent MLP decoders can be combined into a single MLP through appropriate parameter initialization and freezing, Rad-NeRF can obtain further efficiency improvements and approach the efficiency of scaling the MLP width.
>
> **R2-Q3:  Does the sub-NeRF cease to update or contribute to the computation if it receives very low soft gating scores?**
>
> In the Rad-NeRF training framework, even if a sub-NeRF is assigned an extremely low gating score for a particular ray, it still contributes to the rendering of the scene and remains effectively updated.
>
> Firstly, the gating score represents each sub-NeRF's preference for different rays. Although a sub-NeRF is assigned a low score on a certain ray, it still has preferences and contributes to other rays.
>
> Secondly, we implement CV balancing regularization in the training of Rad-NeRF. This regularization prevents the gating module from collapsing onto a specific sub-NeRF while maintaining the different specialties of each sub-NeRF. Consequently, it is rare for a sub-NeRF to contribute extremely little to the total scene during training.
>
> Thirdly, depth-based mutual learning is proposed as a geometry regularization to keep sub-NeRFs learning from each other. Even if a sub-NeRF is assigned a low gating score and learns little from the ground truth color value, it still learns from the fused depth value and remains effectively updated.
>
> **R2-Q4: Are there any automated methods or heuristics for selecting the optimal number of sub-NeRFs based on scene complexity?**
>
> We appreciate the reviewer for the insightful and forward-looking question. In this work, considering the limited computing resources, we mainly adopt the default configuration of two sub-NeRFs and extend it to four sub-NeRFs in the scalability study. Intuitively, more complex and larger scenes require more sub-NeRFs, and the rendering performance of Rad-NeRF will gradually saturate as the number of sub-NeRFs increases. We also conduct additional scalability experiments on MaskTAT, a less complex dataset. As the results show,  the rendering quality saturates when using three sub-NeRFs,  which is similar to the phenomenon observed on the ScanNet dataset, where performance saturation occurs at four sub-NeRFs.
>
> | Method   | Metric   | Baseline   | Size2  | Size3  | Size4  |
> |:--------:|:--------:|:------:|:------:|:------:|:------:|
> | Rad-NeRF | PSNR     | 28.752 | 29.774 | 29.934 | 29.993 |
> |          |     SSIM | 0.914  | 0.920   | 0.925  | 0.924  |
> |          | LPIPS    | 0.140   | 0.130   | 0.125  | 0.124  |
>
> Under such circumstances, determining the optimal number of sub-NeRFs in the Rad-NeRF training framework is an important problem, which we also point out in the conclusion and outlook of the main paper. A possible approach is to adjust the number of sub-NeRFs dynamically during the training process. By setting an appropriate score threshold, sub-NeRFs with average scores lower than the threshold will be removed. Besides, if most sub-NeRFs do not show a significant preference, additional sub-NeRFs can be added.
>
> **R2-Q5: The visualization of the gating score of different sub-NeRFs for the same viewpoint or in scenarios without occlusions.**
>
> In the visualization results, we adopt two sub-NeRFs in all scenes of the TAT dataset. With this setting, the two sub-NeRFs exhibit complementary gating scores for the same view and we omitted the visualization of sub-NeRF2 for brevity in the main paper. We supplement the visualization results of the other sub-NeRF in Figure R.4 (Global Response).
>
> We further provide the gating score visualization in a less-occluded outdoor scene (Playground scene of TAT dataset). As shown in Figure R.4, when rendering in such an open scene with fewer occlusions, the gating score exhibits different characteristic and smooth transition according to the ray directions. This visualization further validates our analysis that as a 4-layer MLP without sinusoidal position encoding, the gating module incorporates smoothness prior implicitly. For unseen viewpoints, especially in less-occluded outdoor scenes, the gating module exhibits smooth and close scores to the nearest seen view.

---

> > ### Comment · Area_Chair_Tsb1 · 2024-08-12
> >
> > Dear Reviewer tZHj:
> >
> > Thanks for reviewing this work. Would you mind to check authors' feedback and see if it resolves your concerns or you may have further comments?
> >
> > Best,
> > AC

---

> > ### Comment · Reviewer_tZHj · 2024-08-12
> >
> > Thank you for your effort. All of my concerns have been addressed. Although better performance is achieved, the paper lacks theoretical analysis, and the proposed architecture is an incremental work by combining the NGP with 'Wang et al. Neural Implicit Dictionary Learning via Mixture-of-Expert Training. ICML 2022'. Additionally, because the design of the hash-table is significantly more important than the subsequent network (as seen in 'Zhu et al. Disorder-invariant Implicit Neural Representation. TPAMI 2024'), it represents a minor innovation by changing one network in the original NGP to two networks (although the paper claims multiple networks, only two are used in experiments). Consequently, I will change my rating from '6 weak accept' to '4 borderline reject'.

---

> > > ### Author Response · Authors · 2024-08-13
> > > **Thanks for the feedback and further discussion from the authors**
> > >
> > > Thanks for the feedback and we're glad our rebuttal has addressed all the concerns! Regarding the new discussion, we'd like to seize this opportunity to express our perspectives and want to reach a consensus with the reviewer.
> > >
> > > ---
> > > > The paper lacks theoretical analysis.
> > >
> > > We acknowledge that our method doesn't have theoretical analysis, we will add this as a discussion of limitations and future work. Nevertheless, the empirical effectiveness of our method is verified very broadly, including quantitive improvements across multiple datasets, comparison with many baselines (single/multi-NeRF frameworks), plugging into existing state-of-the-art frameworks (ZipNeRF), as well as demonstrating potential in scalability study (the performance-parameter scalability curve is better than other scaling dimensions). Moreover, we also have plenty of oracle experiments, ablation studies, and qualitative results analysis that support the logic behind our method design. We think our current empirical evaluation has established our proposed techniques as a worth-considering technique to try out and a new dimension to scale up the model for complex occluded scenes.
> > >
> > > > The proposed architecture is an incremental work by combining the NGP with 'Wang et al. Neural Implicit Dictionary Learning via Mixture-of-Expert Training. ICML 2022'.
> > >
> > > Thanks for recommending this work. RadNeRF is distinct from a combination of Wang et al. and NGP in the following aspects: (1) the targeting scenarios and issues, (2) the design logic and the concrete design of the experts and gating module.
> > > 1. Wang et al. proposed a multi-expert framework to tackle **the generalizable scene modeling problem**, i.e., how to effectively exploit the knowledge of training scenes to model unseen scenes with few-shot views. In contrast, Rad-NeRF focuses on mitigating **the training interference issue arising from inaccurate sampling when modeling one scene**, this issue is especially relevant when this scene is complex and occluded.
> > > 2. Different experts in Wang et al. serve as the basis to construct the implicit field of **different scenes**. In contrast, different experts in Rad-NeRF serve as the basis for constructing the predictions of **different rays**. Therefore, the gating module in Wang et al. takes in the image of the new scene as the input, whereas the gating module in RadNeRF takes in the ray information as the input. **This routing dimension and granularity are vastly different. And we note that the decoupling at the ray dimension and granularity is the key to solving the training interference between the supervision of multiple rays.**
> > >
> > > > Because the design of the hash-table is significantly more important than the subsequent network (as seen in 'Zhu et al. Disorder-invariant Implicit Neural Representation. TPAMI 2024'), it represents a minor innovation by changing one network in the original NGP to two networks (although the paper claims multiple networks, only two are used in experiments).
> > >
> > > Regarding the structure, we agree that hash-table-based representation shows impressive performance. Therefore, we choose to build RadNeRF on this state-of-the-art structure to solidly support RadNeRF's practical usage. Indeed, our method is very simple and introduces very few parameters as it only applies the MoE technique onto the MLP part. Nevertheless, instead of considering this a minor innovation, we truly **regard this simplicity and parameter efficiency as an advantage of our work**, as we have emphasized in our introduction, "RadNeRF is parameter-efficient and super simple to implement".
> > >
> > > Besides, we would like to note that, the "combination" of multiple ray-partitioned experts and the spatial hash-table grid is not a random combination of techniques. Instead, we carefully choose and ablate this choice. Specifically, the hash-table-based feature grid and the consequent MLP both play crucial roles and have distinct effects. The hash grid needs to capture non-ray-related information, and the MLP needs to capture the ray-related information. We find that letting all experts share the spatial feature grid while having different MLPs to predict the ray-related information achieves the best results. As shown by Table 1, Table S.7, we find that the ray-decopuled training of MLPs with a shared spatial feature grid can achieve 21.708, higher than 21.264 achieved by the ray-decoupled training of standalone NGP (note that both of them are better than the baseline NGP 20.722), with much fewer parameters in the meantime.
> > >
> > > Finally, we have experimented with K=2,3,4 experts (Figure 5 and Figure 6 show results on ScanNet, and we add an experiment on MaskTAT in our original reply).
> > >
> > > ---
> > > Thanks again for raising your follow-up concerns to us directly. We hope that our further response can help us reach a new consensus.

---

### Official Review · Reviewer_nJqP · 2024-07-13

**Soundness:** 3
**Presentation:** 3
**Contribution:** 2
**Rating:** 5
**Confidence:** 5

**Summary:**

- The authors decouple the training process of NeRF in the ray dimension and propose a framework where they create an ensemble of sub-NeRFs and train a soft gate module to assign gating scores to these sub-NeRFs based on specific rays.
- The gating module is a 4-layer MLP followed by a softmax function. The gating module takes in ray starting point and direction and outputs a gating score corresponding to each of the sub-NeRFs
- The paper proposes a depth-based loss that compares the fused depth with the depth of the individual NeRFs in order to enhance their robustness.
- Multiple indoor and outdoor scenes from various public datasets were used to show the method's effectiveness.

**Strengths:**

- The paper claims to improve PSNR by 1.5dB with just 0.2% extra parameters.
- The authors have shown the plug-and-play capability of their method by adding it to existing methods like Zip-NeRF.
- While methods like Block-NeRF and Mega-NeRF need a manually defined allocation rule and prior scene knowledge, Rad-NeRF does not require prior scene knowledge.

**Weaknesses:**

- Though the paper claims that their method needs few extra parameters and trains quickly, there only seems to be a slight improvement in quantitative numbers across many scenes.
- The authors have pointed out that the method does not work for few-shot settings.

**Questions:**

- Uncertainty-based methods, like Bayes Rays, consider floaters as  regions corresponding to high uncertainty and remove them by thresholding the scene according to an uncertainty field during rendering. The spatial uncertainty is computed in roughly a minute on any existing method. What is the advantage of Rad-NeRF over such uncertainty-based methods?
- Cross-Ray NeRF (CR-NeRF) leverages interactive information across multiple rays to synthesize occlusion-free novel views with the same appearances as the images. They recover the appearance by fusing global statistics, i.e., feature covariance of the rays and the image appearance. Can the method proposed in this paper solve the problems solved by Rad-NeRF?

**Limitations:**

Yes, the authors have discussed the limitations of the method.

---

> ### Author Rebuttal · Authors · 2024-08-07
>
> We thank the reviewer for the constructive feedback and thoughtful comments. We address the detailed questions and comments below.
>
> **R1-Q1: Performance of Rad-NeRF**
>
> Overall, compared to the Instant-NGP baseline, our scheme shows improvements in the PSNR metric across various datasets: 1.02 on mask-TAT, 0.98 on TAT, 0.57 on 360v2, 0.49 on free dataset and 0.82 on ScanNet dataset. All the improvements are achieved with the default configuration of only two sub-NeRFs (0.02MB extra parameters). Notably, as the number of sub-NeRFs increases, an additional 0.4 PSNR gain is obtained on the ScanNet dataset. We also check the performance of recently published and related work, such as F2-NeRF, which implements a multi-NeRF framework on Instant-NGP. F2-NeRF claims the PSNR metric improvements of 0.15 on 360v2 and 1.91 on the free dataset, which is on par with the Rad-NeRF effect.
>
> Additionally, on the 360v2 dataset, ZipNeRF obtains 0.74 PSNR gain compared to MipNeRF360. Furthermore, we implement the Rad-ZipNeRF framework (a combined version of Rad-NeRF and ZipNeRF) and achieve an additional 0.3 PSNR improvement. The experimental results prove Rad-NeRF's effectiveness and potential for integration with different single-NeRF frameworks.
>
> Finally, Rad-NeRF goes beyond comparison with existing SOTA single-NeRF methods and extends to the comparison with other multi-NeRF training frameworks based on NGP. In these comprehensive evaluations, Rad-NeRF shows superiority over other multi-NeRF training methods.
>
> **R1-Q2: Limitation under the few-shot setting**
>
> Previously, we have included the discussion of the limitation under the few-shot setting. This is because that rendering under the few-shot setting presents a greater challenge to both NeRF's and gating module's generalization ability.
>
> We validate Rad-NeRF's performance in the few-shot setting on the LLFF dataset. For 6/9 training views, Rad-NeRF does not exhibit significant benefits or performance improvements compared to Instant-NGP, with all metrics at the same level. This is because insufficient training data affects the training and generalization of the gating module.
>
> |         |    |  PSNR     |    |      |  SSIM       |  |      |  LPIPS        |  |
> |:-----------:|:------:|:------:|:------:|:------:|:------:|:------:|:------:|:------:|:------:|
> | Method        | 3-view | 6-view | 9-view | 3-view | 6-view | 9-view | 3-view | 6-view | 9-view |
> | Instant-NGP | 16.107 | 19.594 | 21.105 | 0.419  | 0.592  | 0.663  | 0.541  | 0.394  | 0.353  |
> | Rad-NeRF    | 16.626 | 19.351 | 20.979 | 0.452  | 0.595  | 0.661  | 0.506  | 0.399  | 0.344  |
>
> However, when rendering with extremely few training data (3 views), Rad-NeRF achieves significantly better rendering quality. We analyze that when training with very few views, the gating module has minimal impact on NeRF's training. Nonetheless, depth-based mutual learning between multiple sub-NeRFs could still exhibit an effective geometric regularization effect, thereby improving rendering performance. This analysis is also validated by the visualization results in the Global Response. As shown in Figure R.2, compared to the baseline, Rad-NeRF reduces the depth rendering ambiguity and shows better geometry modeling in a 3-view setting.
>
> **R1-Q3: What is the advantage of Rad-NeRF over Bayes Rays?**
>
> As a post-hoc uncertainty assessment framework, Bayes Rays does not change NeRF's training process, only removing "floater" regions corresponding to high uncertainty. However, this solution is not stable and is generally used as an auxiliary solution to improve NeRF's rendering quality.
>
> For example, RobustNeRF treated pixels with larger losses as those with high uncertainty, avoiding the misleading effect of outlier points by discarding the training of those pixels. However, we validated its effect and found that it was difficult to distinguish outlier points from the high-frequency areas that should be learned. Moreover, Instant-NGP regards the spatial points with too low density as regions with high uncertainty and filters these regions when rendering. Although this method works well, it still cannot completely eliminate floaters in difficult scenes and may remove correct regions.
>
> Different from uncertainty-based methods, the proposed Rad-NeRF improves rendering quality by tackling the training interference issue. The depth-based mutual learning method also acts as a geometric regularization to reduce rendering defects. Importantly, Rad-NeRF is essentially orthogonal to these post-training uncertainty removal-based methods and can be integrated with Bayes Rays to obtain further performance improvement. We will incorporate the discussion into the revision.
>
> **R1-Q4: Can the method proposed in CR-NeRF solve the problems solved by Rad-NeRF?**
>
> CR-NeRF works similarly to style transfer/matching NeRF methods, which are primarily proposed to address the challenges of dynamic appearance caused by different capture times and camera settings. Its goal is to control the style/hue of the rendered image from unconstrained image collections. Differently, Rad-NeRF is proposed to tackle the training interference issue in complex scene rendering and improve the rendering performance in normally shot complex scenes.
>
> CR-NeRF proposes to leverage rays' information for modeling appearance, which is an extension and improvement of the appearance modeling scheme in classical NeRF-W. We used to validate the performance of NeRF-W's appearance modeling method on the issue of floaters before and found nearly no improvement. This is because the problem of floaters and other rendering defects in complex scenes is mainly caused by training interference and inaccurate geometry modeling rather than appearance inconsistency. Compared to CR-NeRF, the proposed Rad-NeRF is more suitable to tackle such challenges.

---

> > ### Comment · Reviewer_nJqP · 2024-08-12
> >
> > Thanks for the detailed comments. Most of my concerns are addressed. Is the table for sparse view results in **R1-Q2** for one scene of LLFF or all scenes ?

---

> > > ### Author Response · Authors · 2024-08-12
> > >
> > > Thanks for your feedback and we're happy our rebuttal has addressed most of the concerns!
> > > The quantitative results under few-shot settings in R1-Q2 are the average results on all scenes of the LLFF dataset.
> > >
> > > We're open to any further discussions!

---

> > > ### Author Response · Authors · 2024-08-14
> > > **Final confirmation from the authors**
> > >
> > > Dear Reviewer nJqP,
> > >
> > >   As it's near the end of the discussion period. We would like to take this opportunity to check besides the previous question, if there are any remaining concerns. If our rebuttal has addressed all of your concerns, will you reconsider the overall judgment of our paper to be more positive?
> > >
> > > Thanks very much!
> > >
> > >  Authors

---

### Author Rebuttal · Authors · 2024-08-07

Dear All,

We appreciate all the reviewers' time and efforts invested in reviewing our paper. We are encouraged that the reviewers recognize the effectiveness and scalability(nJqP,tZHj,PUvY,5G52), flexibility and compatibility with different neural rendering approaches (nJqP, tZHj), insightful motivation and idea(tZHj,PUvY), effectiveness of depth-based mutual learning (tZHj,5G52), and good presentation (nJqP,tZHj,PUvY,5G52). We are also thankful for all the concerns and suggestions. The concerns and suggestions are helpful, inspiring, and worth further discussion.

We have responded to each of the questions and suggestions carefully. The related supplementary experiments and figures are supplemented to the PDF file in the 'Global Response' and the specific contents are as below:
1. Figure R.1 shows the scalability study of training time and inference speed.
2. Figure R.2 shows the qualitative comparison results under the 3-view setting;
3. Figure R.3 and Figure R.4 supplement some additional visualizations of gating scores;
4. Figure R.5 shows the independent and fused rendering results of sub-NeRFs.

---

### Decision · Program_Chairs · 2024-09-25

**Decision:**

Accept (poster)

**Comment:**

This paper was reviewed by four experts in the field. The initial reviews were mixed, as reviewers are concerned about the validation of the experiment. However, all these concerns were resolved during the rebuttal session, and all reviewers agreed to accept this work. The reviewers appreciate the novelty of the proposed approach, clear presentation, solid experiments, as well as its effectiveness. AC also agrees that this is well-presented and important work and thus would suggest acceptance.

We recommended the authors to carefully read all reviewers’ final feedback, and revise the manuscript as suggested in the final camera-ready version. We congratulate the authors on the acceptance of their paper!